# Axion Mie theory of electron energy loss spectroscopy in topological insulators

Johannes Schultz[1*], Flavio S. Nogueira[2], Bernd Büchner[1,3,5],
Jeroen van den Brink[2,4,5] and Axel Lubk[1,3]

**1** IFW Dresden, Helmholtzstraße 20, 01069 Dresden, Germany
**2** Institute for Theoretical Solid State Physics, IFW Dresden,
Helmholtzstraße 20, 01069 Dresden, Germany
**3** Department of Physics, TU Dresden, 01069 Dresden, Germany
**4** Institute for Theoretical Physics, TU Dresden, 01069 Dresden, Germany
**5** Würzburg-Dresden Cluster of Excellence ct.qmat

⋆ j.schultz@ifw-dresden.de

## Abstract

Electronic topological states of matter exhibit novel types of responses to electromagnetic fields. The response of strong topological insulators, for instance, is characterized by a so-called axion term in the electromagnetic Lagrangian which is ultimately due to the presence of topological surface states. Here we develop the axion Mie theory for the electromagnetic response of spherical particles including arbitrary sources of fields, i.e., charge and current distributions. We derive an axion induced mixing of transverse magnetic and transverse electric modes which are experimentally detectable through small induced rotations of the field vectors. Our results extend upon previous analyses of the problem. Our main focus is on the experimentally relevant problem of electron energy loss spectroscopy in topological insulators, a technique that has so far not yet been used to detect the axion electromagnetic response in these materials.

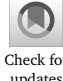
# 1 Introduction

It has been long known theoretically that in CP-violating theories topological defects become electrically charged, one of the most prominent examples being the induced fractional electric charge in 't Hooft-Polyakov monopoles [1]. This result follows immediately from the presence of a CP-violating interaction in the Lagrangian of the gauge theory,

$$\mathcal{L}_a = \frac{\Theta g^2}{32\pi^2} \epsilon_{\mu\nu\lambda\rho} F^{\mu\nu} F^{\lambda\rho} \,, \tag{1}$$

where $F_{\mu\nu}$ is the usual field strength tensor of the gauge sector of the theory and $g$ is the gauge coupling. The significance of the parameter $\Theta$ varies depending on the context, but we will refer to it here simply as the *axion*, in reference to the chiral (or axial) anomaly [2]. Within an Abelian gauge theory context, observable consequences of axion electrodynamics have been discussed for some time, with references to materials already being made in early papers. For instance, Nielsen and Ninomiya [3] mentioned HgCdTe and Wilczek [4] discusses an application of axion electrodynamics to PbTe, following a suggestion by Fradkin *et al.* [5]. However, it was not until more recently with the prediction and discovery of several topological materials, notably topological insulators (TIs) [6,7] and Weyl semimetals [8], that the many possible interesting experimental consequences of axion electrodynamics came closer to spotlight. In this work we mainly focus on TIs, but the theory described here applies with minor modifications to Weyl semimetals as well.

Three-dimensional TIs feature a quantum Hall electromagnetic response at the surface, which emerges as a consequence of an insulator behavior in the bulk characterized effectively by the following axion electrodynamics Lagrangian [9],

$$\mathcal{L} = \frac{1}{8\pi}(\epsilon E^2 - B^2) - \frac{\alpha\Theta}{4\pi^2} \boldsymbol{E} \cdot \boldsymbol{B} \,, \tag{2}$$

where $\boldsymbol{E}$ and $\boldsymbol{B}$ are the electric and magnetic field, respectively, $\alpha = e^2/(\hbar c)$ is the fine-structure constant and we have assumed a paramagnetic bulk with magnetic permeability $\mu = 1$. For TIs where either time-reversal or inversion symmetry holds, $\Theta = \pi$. Thus, a TI sample in vacuum may be considered as the problem of a topological dielectric satisfying the Maxwell equations,

$$\boldsymbol{\nabla} \cdot \left( \epsilon \boldsymbol{E} - \frac{\alpha\Theta}{\pi} \boldsymbol{B} \right) = 4\pi\rho \,, \tag{3}$$

$$\boldsymbol{\nabla} \times \left( \boldsymbol{B} + \frac{\alpha\Theta}{\pi} \boldsymbol{E} \right) = ik \left( \epsilon \boldsymbol{E} - \frac{\alpha\Theta}{\pi} \boldsymbol{B} \right) + \frac{4\pi}{c} \boldsymbol{j} \,, \tag{4}$$

$$\boldsymbol{\nabla} \times \boldsymbol{E} + ik\boldsymbol{B} = 0 \,, \tag{5}$$

$$\boldsymbol{\nabla} \cdot \boldsymbol{B} = 0 \,, \tag{6}$$

where we have used $\boldsymbol{E} \to \boldsymbol{E} e^{-ickt}$, $\boldsymbol{B} \to \boldsymbol{B} e^{-ickt}$, current $\boldsymbol{j} \to \boldsymbol{j} e^{-ickt}$, charge density $\rho \to \rho e^{-ickt}$, and assumed a frequency-dependent dielectric function, $\epsilon(\omega, \boldsymbol{r}) = \epsilon(ck, \boldsymbol{r})$. The resulting constitutive relations then read,

$$\boldsymbol{D} = \epsilon \boldsymbol{E} - \frac{\alpha\Theta}{\pi} \boldsymbol{B} \,, \tag{7}$$

$$\boldsymbol{H} = \boldsymbol{B} + \frac{\alpha\Theta}{\pi} \boldsymbol{E} \,. \tag{8}$$

When $j = 0$ the above constitutive equations can be rewritten as,

$$D = \epsilon E + \frac{\alpha\Theta}{i\pi k}\nabla \times E\,, \tag{9}$$

$$B = \mu H - \frac{\alpha\Theta}{i\pi k}\mu\nabla \times H\,, \tag{10}$$

where $\mu = \epsilon/[\epsilon + (\alpha\Theta/\pi)^2]$. As noted in Ref. [10], in this case the problem reduces to the one of light scattering by an optically active sphere, solved by Bohren long time ago [11, 12]. A closely related problem arises also in the study of a so called chiral dielectric-magnetic medium [13].

Using Eq. (5), we can rewrite Eq. (4) as,

$$\nabla \times B = \frac{4\pi}{c}j - \frac{\alpha}{\pi}\nabla\Theta \times E + ik\epsilon E\,. \tag{11}$$

Note that since $\Theta$ is uniform inside the TI and vanishes outside it, $\nabla\Theta$ is proportional to a delta function on the TI surface, $\Sigma_{TI}$. From the above equation we can easily read off the surface quantum Hall current,

$$j_H(r) = \frac{e^2\Theta}{2\pi h}\int dS(u,v) \times E(r)\delta^3(r - r_S(u,v))\,, \tag{12}$$

where $dS(u,v) = dudv[\partial r_S(u,v)/\partial u \times \partial r_S(u,v)/\partial v]$ and $r_S(u,v) \in \Sigma_{TI}$, with $(u,v) \in \mathbb{R}^2$ being associated to a parametrization of $\Sigma_{TI}$. We have that $\sigma_H = e^2\Theta/(2\pi h)$ is the Hall conductivity. For $\Theta = \pi$ or, more generally, $\Theta = 2\pi(n+1/2)$, $n \in \mathbb{Z}$, we obtain a half-quantized Hall conductivity [9]. Note that a non-vanishing Hall conductivity is equivalent to a non-null surface admittance employed in other studies to describe the conducting surface states of a TI [10].

Given the possibility of a more general quantization profile for $\Theta$, it is natural to wonder whether values significantly larger than the canonical $\Theta = \pi$ one are really possible. This would enhance the sensitivity of experimental probes, since this would allow to compensate against factors of $\alpha$, especially in those experimental responses leading to effects $\sim \mathcal{O}(\alpha^2)$ However, values of $\Theta$ beyond $n = 0$ are not only experimentally unrealistic, they can actually be excluded at a more fundamental level. Since very large values of $\Theta$ ($\sim 1000$) have been considered in the recent literature [10, 14], let us briefly comment on this point by recalling some well known facts related to the parity anomaly [15] in this context [7, 9].

First we note that the axion term in the Lagrangian (2) can be expressed in terms of a total derivative of a Chern-Simons term, which is more easily seen within a covariant formulation as given in Eq. (1), which can be rewritten as,

$$\mathcal{L}_a = \frac{\alpha\Theta}{8\pi^2}\partial^\mu(\epsilon_{\mu\nu\lambda\rho}A^\nu\partial^\lambda A^\rho)\,, \tag{13}$$

where we have replaced $g^2$ by $\alpha$. Hence, assuming for simplicity a flat TI surface perpendicular to the $z$-axis separating the TI bulk from vacuum, we obtain the following surface Chern-Simons (CS) contribution to the action,

$$S_{CS} = \frac{\alpha\Theta}{8\pi^2}\epsilon_{\mu\nu\lambda}A^\mu\partial^\nu A^\lambda\,. \tag{14}$$

On the other hand, it is well known that integrating out gapped Dirac fermions in 2+1 dimensions coupled to an electromagnetic field generate the CS action above with $\Theta = \pi$, a result

reflecting the so-called parity anomaly [15,16]. Precisely the same gapped Dirac modes arise on a TI surface when time-reversal (TR) symmetry is broken. Therefore, other values of $\Theta$, especially ones significantly larger than $\pi$, seem to be excluded on the basis of this argument. As a final remark concerning the value of $\Theta$ in the bulk when TR is broken, recall that $\Theta = \pi$ is still enforced provided inversion symmetry is not broken [9].

In the past several years a number of predictions of electromagnetic phenomena related to TIs have been made based on the above equations of axion electrodynamics [9,17–23]. Experimentally, such phenomena are often difficult to probe, since some effects manifest themselves only through corrections $\sim \mathcal{O}(\alpha^2)$. There are effects occurring at $\mathcal{O}(\alpha)$, which are more accessible to experimental probes. This is precisely the case with, for instance, the Faraday and Kerr rotation in TIs [9,18], which has been measured using $Bi_2Se_3$ [24] and strained HgTe [25] samples. The derivation of the effect is elementary and follows directly from the Maxwell equations above with the boundary conditions modified by the discontinuity of $\Theta$ across the TI surface [9]. The experimental verification of this result using samples of inversion symmetric, epitaxially grown $Bi_2Se_3$ [24] showed the robustness of the value $\Theta = \pi$, as we have discussed above.

Another $\mathcal{O}(\alpha)$ effect that has been predicted in the literature concerns surface plasmon polaritons (SPPs) on the planar surface of a TI [19]. In this work the standard analysis of SPPs on a planar geometry has been extended to TIs in the same geometry. As a matter of fact, the same Maxwell equations can be used, except for a nontrivial change in the boundary conditions due to the presence of the axion term in Eq. (2) [19]. Indeed, the TI surface introduces a discontinuity in $\Theta$ mixing electric and magnetic boundary conditions even in the static case [17,21,23]. As a consequence, while there is only a transverse magnetic and no transverse electric component of the electric field in the usual theory for SPPs on planar surfaces, such a component does not vanish in the TI case.

One of the most important effects of light scattering beyond the planar geometry originates from the so called Mie theory [26]. This theory deals with plane wave scattering by a dielectric sphere and amounts to solving the Maxwell equations with appropriate boundary conditions [27,28]. Developing the *axion Mie* theory is severely complicated by the disparity between the spherical symmetry of the target relative to the incoming plane waves; including the axion modifies the boundary conditions relative to the standard Mie theory calculations. Even if certain aspects of the Mie theory for spherical TIs has been considered recently [14], there are many relevant issues that remain to be resolved. This includes an accurate treatment of the modified boundary conditions beyond the first order perturbation level as well as the incorporation of arbitrary sources of fields (i.e., charges and currents) and the identification of suitable experimental setups allowing to measure the implications of the axion term. Another important distinction between the standard Mie theory for a dielectric sphere and the corresponding extension to the spherical TI case is that the latter has a metallic surface where induced (surface) currents feature electrons with the spin-momentum locking property. The latter is actually responsible for the peculiar electromagnetic response whose content is captured by the Lagrangian (Eq. (2)).

In the following we elaborate on these general considerations restricting ourselves to spherical symmetry. This allows to derive analytical solutions for the electromagnetic response of TI spheres, notably including Hall currents, magnetic fluxes, and scattering cross-sections. Our considerations essentially amount to an extension of classical Mie theory to include the ramification of the axion term. While the homogeneous case was already treated using different approaches [10,14], we do not restrict ourselves to the charge- and current-free case and allow for arbitrary external sources of fields. In particular, external charges within the simulation volume as present in charge particle scattering experiments such as electron energy loss spectroscopy (EELS) are not covered by previous studies. Note that exploiting spherical symmetry

requires isotropic dielectric functions which is in generally (especially for TIs) not the case. To overcome this constraint the problem needs to be tackled numerically. We will, however, focus somewhat on particular case of localized surface plasmons (LSPs), i.e., negative dielectric response bands. This serves as an archetypal model system for the LSPs on TI nanoparticles of more complicated shape, noting that in particular topological characteristics (i.e. the value of $\Theta$) do not depend on the particular particle shape. For other regimes of positive dielectric response, which may be treated with the same formalism, we refer to the literature.

## 2 Solution of the field equations

In order to describe the dielectric response of a TI the curl of the Maxwell Eq. (4) is computed inserting Eq. (5) to replace the magnetic flux density

$$\boldsymbol{\nabla} \times (\boldsymbol{\nabla} \times \boldsymbol{E}) - k^2 \epsilon \boldsymbol{E} - i k \frac{\alpha}{\pi} \boldsymbol{\nabla}\Theta \times \boldsymbol{E} = -i \frac{4\pi k}{c} \boldsymbol{j} \,. \tag{15}$$

For spatially constant $\epsilon$ and $\Theta$ the standard form holds,

$$\triangle \boldsymbol{E} + k^2 \epsilon \boldsymbol{E} = i \frac{4\pi k}{c} \boldsymbol{j} + \frac{4\pi}{\epsilon} \boldsymbol{\nabla}\rho \,. \tag{16}$$

The general dielectric response of the sphere (i.e. Mie theory) is obtained by solving Eq. (15) exploring the spherical symmetry. In the following we tackle the problem by a piecewise solution in regions of spatially constant $\epsilon$ and $\Theta$ (i.e., within and outside of the sphere) and fixing the missing integration constants through appropriate boundary conditions. Apart from assuming spherical symmetry, there are no approximations in the expressions above and in this spirit we will continue below when presenting the exact solution of the problem. It is furthermore informative to perform a Helmholtz decomposition of the dielectric response equation at this state as it allows to distinguish a longitudinal and transverse part of the solution on this very fundamental level. We will make use of these results further below.

We begin with expanding the electric field into vector spherical harmonics [29–31] (following the definition by Barrera et al. [31]),

$$\mathbf{E}(\boldsymbol{r}) \;=\; \sum_{l=0}^{\infty} \sum_{m=-l}^{l} \left( E_{lm}^{\perp}(r)\mathbf{Y}_{lm} + E_{lm}^{(1)}(r)\boldsymbol{\Psi}_{lm} + E_{lm}^{(2)}(r)\boldsymbol{\Phi}_{lm} \right), \tag{17}$$

which form a complete basis for vector fields in three dimensions and hence give rise to a general representation of electric and magnetic fields adapted to spherical coordinates. That notably includes free vacuum solutions, which are typically considered in the literature of Mie scattering, but also fields occurring in the presence of non-zero charges and currents. While free solutions with $\nabla \cdot \boldsymbol{D} = 0$ everywhere may be represented by a restricted set of two-dimensional vector spherical harmonics, which typically correspond to the poloidal and toroidal fields [10, 12, 27], the general inhomogeneous case requires a complete three-dimensional representation as the one employed in this work. Note furthermore that the $l = 0$ case is peculiar in that both $\boldsymbol{\Psi}$ and $\boldsymbol{\Phi}$ vanish identically (ultimately a consequence of the hairy ball theorem). As a consequence the $l = 0$ modes play a special role throughout.

Taking into account the completeness relations of the vector spherical harmonics Eqs., (15) and (16) reduce to a system of three ordinary differential equations for each vector spherical

harmonics. After setting $E_{lm}^{\perp} = E_{lm}^{(\perp)}/r$ we explicitly obtain

$$
r^2 \frac{\mathrm{d}^2 E_{lm}^{(\perp)}}{\mathrm{d}r^2} + 2r \frac{\mathrm{d}E_{lm}^{(\perp)}}{\mathrm{d}r} + \left(k^2 \epsilon r^2 - l(l+1)\right) E_{lm}^{(\perp)}
$$
$$
= -i \frac{4\pi k}{c} r^3 j_{lm}^{\perp} + \frac{12\pi}{\epsilon} r^3 (\boldsymbol{\nabla}\rho)_{lm}^{(1)} + \frac{4\pi}{\epsilon} r^4 \frac{\mathrm{d}(\boldsymbol{\nabla}\rho)_{lm}^{(1)}}{\mathrm{d}r} \qquad , \tag{18}
$$

for the first component $E^{(\perp)}$. The second component $E^{(1)}$ is directly linked to the first through the first Maxwell equation

$$
E_{lm}^{(1)} = \frac{r}{l(l+1)} \frac{\mathrm{d}E_{lm}^{\perp}}{\mathrm{d}r} + \frac{2}{l(l+1)} E_{lm}^{\perp} - \frac{4\pi}{\epsilon l(l+1)} r^2 (\boldsymbol{\nabla}\rho)_{lm}^{(1)} . \tag{19}
$$

Note that the apparent divergence in case of $l = 0$ dissolves under closer inspection as the $\boldsymbol{\Psi}$ component vanishes. Indeed, $l = 0$ solutions do not occur in the homogeneous case as the previous two differential equations are not compatible in this case. The third differential equation for $E^{(2)}$ has again the same mathematical structure as the first

$$
r^2 \frac{\mathrm{d}^2 E_{lm}^{(2)}}{\mathrm{d}r^2} + 2r \frac{\mathrm{d}E_{lm}^{(2)}}{\mathrm{d}r} + \left(k^2 \epsilon r^2 - l(l+1)\right) E_{lm}^{(2)} = i \frac{4\pi k r^2}{c} j_{lm}^{(2)}. \tag{20}
$$

Both, the first and third equation, are inhomogeneous ordinary second order differential equation in $E_{lm}$ of (modified) spherical Bessel type (after absorbing $k\sqrt{\epsilon} = kn$ into $r$). The fate of the equation being of modified type or not is decided by the sign of of $\epsilon$ (if the latter is real). While in vacuum $\epsilon$ is positive corresponding to a spherical Bessel equation, within the sphere both positive and negative sign (depending on the frequency) can occur. The general solution, notably including whispering gallery modes when $\epsilon_{\mathrm{sphere}} > \epsilon_{\mathrm{vac}}$ (positive sign of $\epsilon$) as well as evanescent excitations i.e., surface plasmons (negative sign of $\epsilon$) leads to Bessel functions of complex arguments. Note furthermore that the solution space can be further restricted to those, which do not diverge at the origin.

The differential equations above do not contain any modifications due to the topological term. Indeed, being a divergence term, the axion coupling only affects the boundary conditions applying at the interface between vacuum and TI sphere, as anticipated. These internal boundary conditions are derived from partial derivatives normal to the surface inserting Maxwell's equations as usual.

From the homogeneous Maxwell equations (second and third) we obtain

$$
\frac{r\,\mathrm{d}E_{lm1}^{\perp}}{\mathrm{d}r} + 2E_{lm1}^{\perp} = \frac{r\,\mathrm{d}E_{lm2}^{\perp}}{\mathrm{d}r} + 2E_{lm2}^{\perp} \tag{21}
$$

and

$$
-\frac{l(l+1)}{r} \left(E_{lm1}^{(2)} - E_{lm2}^{(2)}\right) = 0 . \tag{22}
$$

The BCs obtained from the inhomogeneous Maxwell equations (first and fourth) read

$$
E_{lm2}^{\perp} - n^2 E_{lm1}^{\perp} = \frac{i\alpha}{k\pi} \frac{l(l+1)}{r} E_{lm}^{(2)} \Theta \tag{23}
$$

and

$$
-\frac{\alpha}{\pi l(l+1)} \left(r \frac{\mathrm{d}E_{lm}^{\perp}}{\mathrm{d}r} + 2E_{lm}^{\perp}\right) \Theta = \frac{1}{ik} \left(\frac{\mathrm{d}E_{lm1}^{(2)}}{\mathrm{d}r} - \frac{\mathrm{d}E_{lm2}^{(2)}}{\mathrm{d}r}\right) . \tag{24}
$$

Similarly to the field equations only 2 of the 4 BCs are affected by the topological term (i.e., those pertaining to the inhomogeneous Maxwell equations).

## Plasmonic Regime

The solutions of the differential Eqs. (18) - (20) allows to treat several problems e.g., scattering of electromagnetic waves or the electromagnetic response of a sphere to external charges and currents. However, we focus on plasmonic excitations which can be excited by plane waves or other external sources. The fundamental condition for plasmonic behavior is an opposite sign of the dielectric function of the nanoparticle with respect to the surrounding medium. As a consequence, we analyze the case of negative dielectric function of the nanoparticle, since we assume the vacuum ($\epsilon = 1$) as a surrounding medium. This condition typically holds for optical and near-infrared frequencies. Representatives of plasmonic TIs at optical frequencies are $Bi_2Se_3$ [32], [33] and $Bi_2Te_3$ [34].

## Homogeneous Case

The general solutions of the above differential equations split into a homogeneous and inhomogeneous part. The homogeneous solution, which includes the important problem of light scattering on a sphere (i.e. Mie scattering), is considered first (for details see Appendix A.1). Even though the homogeneous axion Mie problem has already been treated comprehensively [10, 14], it is worth to revisit it here, since we use an alternative set of vector spherical harmonics [31] that is more suitable to solve the full inhomogeneous problem (including arbitrary external charges and currents), and introduce the scattering matrix formalism employed later on.

General solutions for the above spherical Bessel differential equations may be generally expanded into spherical Bessel functions $j_l$ and $y_l$ (first and second kind, respectively). For the radial component (Eq. (18)) this expansion explicitly reads

$$E_{lm}^{(\perp)}(r) = \begin{cases} a_l^{(\perp)} j_l^i(nkr), & r < R \\ b_l^{(\perp)} j_l^o(kr) + c_l^{(\perp)} y_l(kr), & r \geq R. \end{cases} \tag{25}$$

Here the indices i and o correspond to the inner and outer areas of the sphere. Note that the modified Bessel function of second kind does not appear in the interior of the sphere as it diverges at the origin. From Eq. (19) the first tangential component can be derived

$$E_{lm}^{(1)}(r) = \begin{cases} a_l^{(\perp)} \left[ \frac{nk}{l(l+1)} j_{l+1}^i(nkr) + \frac{1}{l} \frac{j_l^i(nkr)}{r} \right], & r < R \\ b_l^{(\perp)} \frac{rkj_{l-1}^o(kr) + lj_l^o(kr)}{rl(l+1)} + c_l^{(\perp)} \frac{rky_{l-1}(kr) + ly_l(kr)}{rl(l+1)}, & r \geq R. \end{cases} \tag{26}$$

The solution to Eq. (20) for the second tangential component reads

$$E_{lm}^{(2)}(r) = \begin{cases} a_l^{(2)} j_l^i(nkr), & r < R \\ b_l^{(2)} j_l^o(kr) + c_l^{(2)} y_l(kr), & r \geq R. \end{cases} \tag{27}$$

Again, we observe that the radial and first tangential component form a coupled solution, whereas the second tangential component is independent. In the first case the magnetic field (i.e., the curl of the electric field) is strictly tangential to the sphere, whereas in the second case the electric field is tangential. Following the notation of [35] we will refer to the latter as magnetic (m) and the former as electric (e) modes here. Keep in mind that we set $E_{lm}^{\perp} = E_{lm}^{(\perp)}/r$ in order to have less headaches solving Eq. (18). For further calculations we need to transform $E_{lm}^{(\perp)}$ and in consequence $a_{lm}^{(\perp)}$, $b_{lm}^{(\perp)}$ and $c_{lm}^{(\perp)}$ back to $E_{lm}^{\perp}$, $a_{lm}^{\perp}$, $b_{lm}^{\perp}$ and $c_{lm}^{\perp}$. Indeed, the coupled radial and first tangential component form the poloidal component of the electrical field (more precisely: of the the electric displacement for which $\nabla \cdot \boldsymbol{D} = 0$ everywhere), whereas the second tangential component is the toroidal field. Both modes are completely decoupled

(and orthogonal), which is exploited in the classical Mie solution by expanding the fields directly in poloidal and toroidal fields from the outset. Note furthermore that in both cases the homogeneous solutions are degenerate with respect to $m$.

In order to determine the six expansion coefficients (remember that $E^{(1)}$ can be computed once $E^{(\perp)}$ is known) of the homogeneous solutions a corresponding number of boundary conditions has to be provided. Four of them stem from the internal boundaries at the surface of the sphere, noted previously. The remaining two are given either by normalization conditions or external boundary conditions. Indeed, it is only at this stage, where deviations from the classical Mie theory are introduced by the axion term, as anticipated. We first consider the impact of the topological term on the so-called normal modes, i.e., modes decaying to zero at infinity. They are obtained by solving the system of 4 internal BCs (Eq. system (79) for the 4 coefficients $b_l$, $c_l$ and fixing the $a_l$ by normalization conditions (see Appendix A.2):

$$b_l^{(2)} = \frac{a_l^{(2)}\left(y_l \frac{dj_l^i}{dr} - j_l^i \frac{dy_l}{dr}\right)}{y_l \frac{dj_l^o}{dr} - j_l^o \frac{dy_l}{dr}} + \frac{a_l^\perp \frac{i\alpha\Theta kR}{\pi l(l+1)} \frac{d(rj_l^i)}{rdr} y_l}{y_l \frac{dj_l^o}{dr} - j_l^o \frac{dy_l}{dr}}, \tag{28}$$

$$c_l^{(2)} = \frac{a_l^{(2)}\left(j_l^o \frac{dj_l^i}{dr} - j_l^i \frac{dj_l^o}{dr}\right)}{j_l^o \frac{dy_l}{dr} - y_l \frac{dj_l^o}{dr}} + \frac{a_l^\perp \frac{i\alpha\Theta kR}{\pi l(l+1)} \frac{d(rj_l^i)}{rdr} j_l^o}{j_l^o \frac{dy_l}{dr} - y_l \frac{dj_l^o}{dr}}, \tag{29}$$

$$b_l^\perp = \frac{a_l^\perp\left(y_l \frac{d(rj_l^i)}{rdr} - n^2 j_l^i \frac{d(ry_l)}{rdr}\right)}{y_l \frac{dj_l^o}{dr} - j_l^o \frac{dy_l}{dr}} + \frac{a_l^{(2)} \frac{\alpha l(l+1)\Theta}{i\pi kR} \frac{d(ry_l)}{rdr} j_l^i}{y_l \frac{dj_l^o}{dr} - j_l^o \frac{dy_l}{dr}}, \tag{30}$$

$$c_l^\perp = \frac{a_l^\perp\left(j_l^o \frac{d(rj_l^i)}{rdr} - n^2 j_l^i \frac{d(rj_l^o)}{rdr}\right)}{j_l^o \frac{dy_l}{dr} - y_l \frac{dj_l^o}{dr}} + \frac{a_l^{(2)} \frac{\alpha l(l+1)\Theta}{i\pi kR} \frac{d(rj_l^o)}{rdr} j_l^i}{j_l^o \frac{dy_l}{dr} - y_l \frac{dj_l^o}{dr}}. \tag{31}$$

From the equations above it is clear that indeed the modifications due to the axion term amount to a mixing of the originally decoupled electric and magnetic modes, which has been previously noted for the half-plane boundary case by Karch [19]. Noting that fields of electric and magnetic modes are perpendicular, their weak mixing is equivalent to a rotation of the electric or magnetic field vectors linear in $\alpha$, which has been successfully detected in the half plane geometry and may be also amenable to experimental detection in case of the sphere.

Fig. 1 shows a comparison between classical electric and magnetic normal modes for $l = 1$ and the corresponding TI solutions (normalized by setting $a_l^\perp$ or $a_l^{(2)}$ to zero, respectively). In the topologically trivial setting (left columns of Fig. 1 respectively) we readily observe that electric and magnetic normal modes are completely decoupled. In case of the TI sphere on the right columns of Fig. 1 respectively, on the other hand, originally purely electric and magnetic modes mix due to the axion term, i.e., they acquire a small magnetic mode and electric mode character respectively and become hybrid modes. Similarly, the 3D representation of the $l = 1$, $m = 1$ modes (Fig. 2) reveals a tilting out of the tangential plane of the B-field (E-field) in case of the electric (magnetic) mode, which corresponds to the above noted mixing of electric and magnetic modes.

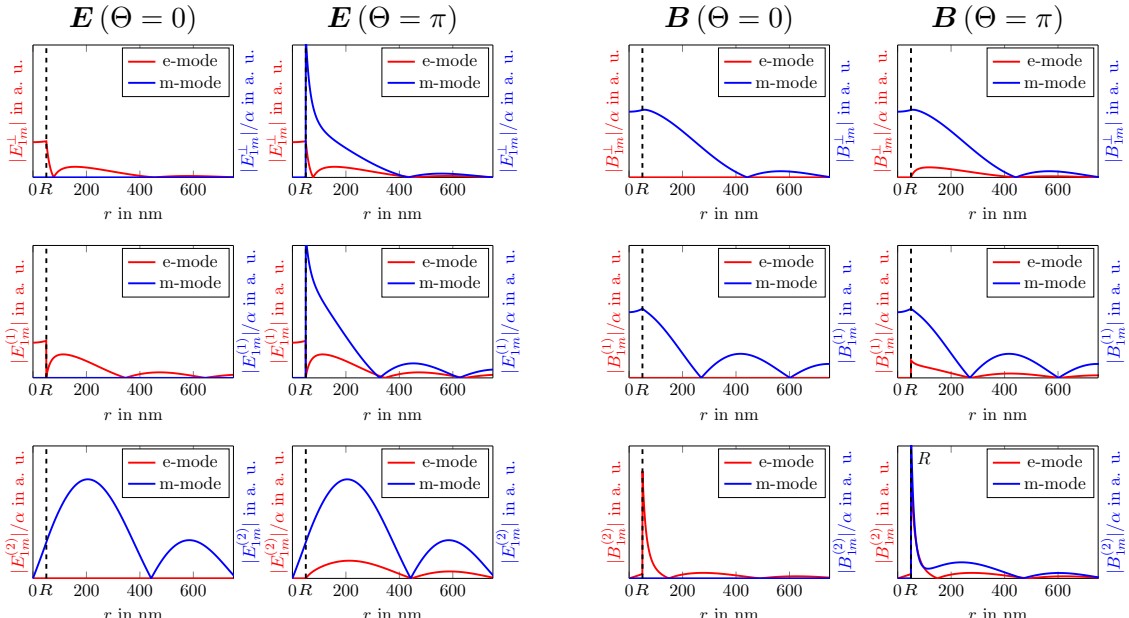

Figure 1: Radial dependency of electric (Eq. (25 - 27)) and magnetic (derived using Eq. (5)) field components (absolute value) of $l = 1$ normal modes for topologically trivial ($R = 50$ nm, $\epsilon = -1$, $\Theta = 0$) and TI sphere ($R = 50$ nm, $\epsilon = -1$, $\Theta = \pi$) embedded in vacuum respectively at $\hbar\omega = 2$ eV. Note that the y-axis was always scaled in the same arbitrary units.

More insight into the origin of that behavior may be obtained from the analysis of the Hall charges (Eq. 23) and Hall currents (Eq. (12) and Eq.(24)) associated to the axion term. Fig. 3 shows the Hall charge and current of the $l = 1, m = 1$ electric mode. Accordingly, the Hall current has a source and sink due to oscillating charges and produces magnetic fields with components normal to the surface, i.e. a magnetic mode. Moreover, the number of sources and sinks increases with the mode order, e.g., a quadrupolar structure is visible for the $l = 2, m = 2$ electric mode (see Fig. 3).

Slightly reformulating the dependencies between the expansion coefficients and fixing the missing 2 BCs by an external excitation allows the computation of any electromagnetic response of the TI in the absence to charges and currents, i.e., the description of (resonant) photon scattering. As an example we note the solution to the scattering of a linearly polarized plane wave. Note that due to conservation of angular momentum the whole problem separates in $l$. In order to obtain an analytic expression of the scattering and extinction cross section ($\sigma_s$, Eq. (38) and $\sigma_e$, Eq. (39)) we first express the expansion in the vacuum regions in terms of spherical Hankel functions of first and second kind with excitation coefficients $d_l = \frac{b_l - ic_l}{2}$ and $e_l = \frac{b_l + ic_l}{2}$ respectively, which allow a separation of outgoing and incoming spherical waves, respectively. In a second step the outgoing waves are expressed as a linear function of the incoming one (see Eq. system (79)):

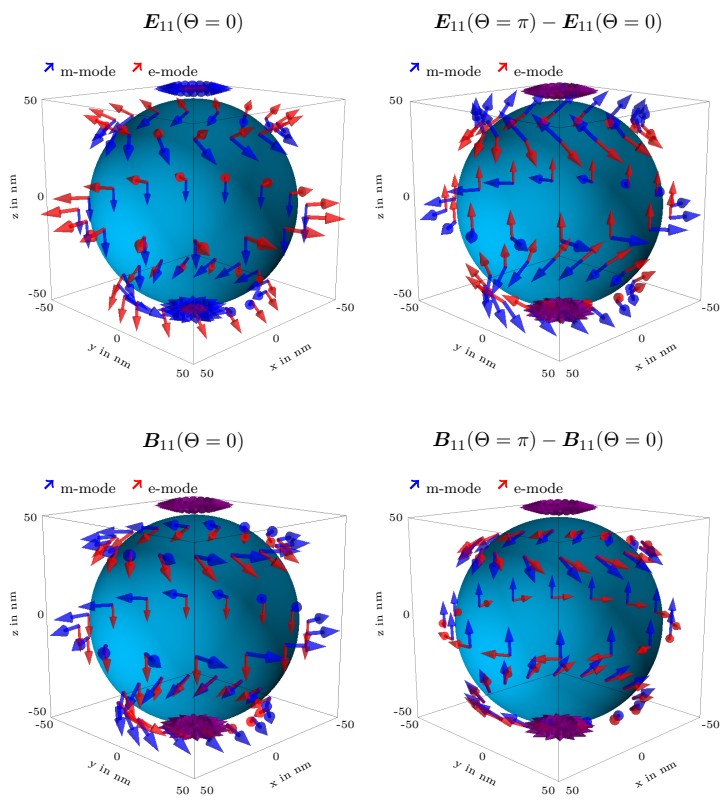

Figure 2: 3D vector representation (based on Eq. (17)) at $r = 55\,\text{nm}$ of the $l = 1, m = 1$ electric and magnetic modes (absolute value) for the topologically trivial sphere ($R = 50\,\text{nm}, \epsilon = -1$) embedded in vacuum at $\hbar\omega = 2\,\text{eV}$ (left hand side) and their modifications due to the axion term (right hand side). Note, that the corresponding magnetic field can be directly derived from the electric field using the third Maxwell equation (Eq. (5)).

$$
\begin{aligned}
d_l^{\perp} &= -\frac{\left(h_l^{(1)}\frac{\mathrm{d}j_l^{\mathrm{i}}}{\mathrm{d}r} - j_l^{\mathrm{i}}\frac{\mathrm{d}h_l^{(1)}}{\mathrm{d}r}\right)\left(h_l^{(2)}\frac{\mathrm{d}(rj_l^{\mathrm{i}})}{r\mathrm{d}r} - n^2 j_l^{\mathrm{i}}\frac{\mathrm{d}(rh_l^{(2)})}{r\mathrm{d}r}\right) - \frac{\alpha^2\Theta^2}{\pi^2}h_l^{(1)}\frac{\mathrm{d}(rj_l^{\mathrm{i}})}{r\mathrm{d}r}j_l^{\mathrm{i}}\frac{\mathrm{d}(rh_l^{(2)})}{r\mathrm{d}r}}{\left(h_l^{(1)}\frac{\mathrm{d}j_l^{\mathrm{i}}}{\mathrm{d}r} - j_l^{\mathrm{i}}\frac{\mathrm{d}h_l^{(1)}}{\mathrm{d}r}\right)\left(h_l^{(1)}\frac{\mathrm{d}(rj_l^{\mathrm{i}})}{r\mathrm{d}r} - n^2 j_l^{\mathrm{i}}\frac{\mathrm{d}(rh_l^{(1)})}{r\mathrm{d}r}\right) - \frac{\alpha^2\Theta^2}{\pi^2}h_l^{(1)}\frac{\mathrm{d}(rj_l^{\mathrm{i}})}{r\mathrm{d}r}j_l^{\mathrm{i}}\frac{\mathrm{d}(rh_l^{(1)})}{r\mathrm{d}r}}e_l^{\perp} \\
&\quad + 2\frac{\frac{\alpha\Theta l(l+1)}{\pi kR}j_l^{\mathrm{i}}\frac{\mathrm{d}(rj_l^{\mathrm{i}})}{r\mathrm{d}r}\left(j_l^{\mathrm{o}}\frac{\mathrm{d}(ry_l)}{r\mathrm{d}r} - y_l\frac{\mathrm{d}(rj_l^{\mathrm{o}})}{r\mathrm{d}r}\right)}{\left(h_l^{(1)}\frac{\mathrm{d}j_l^{\mathrm{i}}}{\mathrm{d}r} - j_l^{\mathrm{i}}\frac{\mathrm{d}h_l^{(1)}}{\mathrm{d}r}\right)\left(h_l^{(1)}\frac{\mathrm{d}(rj_l^{\mathrm{i}})}{r\mathrm{d}r} - n^2 j_l^{\mathrm{i}}\frac{\mathrm{d}(rh_l^{(1)})}{r\mathrm{d}r}\right) - \frac{\alpha^2\Theta^2}{\pi^2}h_l^{(1)}\frac{\mathrm{d}(rj_l^{\mathrm{i}})}{r\mathrm{d}r}j_l^{\mathrm{i}}\frac{\mathrm{d}(rh_l^{(1)})}{r\mathrm{d}r}}e_l^{(2)}, \quad (32)
\end{aligned}
$$

$$
\begin{aligned}
d_l^{(2)} &= -2\frac{\frac{\alpha\Theta kR}{\pi l(l+1)}j_l^{\mathrm{i}}\frac{\mathrm{d}(rj_l^{\mathrm{i}})}{r\mathrm{d}r}\left(j_l^{\mathrm{o}}\frac{\mathrm{d}y_l}{\mathrm{d}r} - y_l\frac{\mathrm{d}j_l^{\mathrm{o}}}{\mathrm{d}r}\right)}{\left(h_l^{(1)}\frac{\mathrm{d}j_l^{\mathrm{i}}}{\mathrm{d}r} - j_l^{\mathrm{i}}\frac{\mathrm{d}h_l^{(1)}}{\mathrm{d}r}\right)\left(h_l^{(1)}\frac{\mathrm{d}(rj_l^{\mathrm{i}})}{r\mathrm{d}r} - n^2 j_l^{\mathrm{i}}\frac{\mathrm{d}(rh_l^{(1)})}{r\mathrm{d}r}\right) - \frac{\alpha^2\Theta^2}{\pi^2}h_l^{(1)}\frac{\mathrm{d}(rj_l^{\mathrm{i}})}{r\mathrm{d}r}j_l^{\mathrm{i}}\frac{\mathrm{d}(rh_l^{(1)})}{r\mathrm{d}r}}e_l^{\perp} \\
&\quad - \frac{\left(h_l^{(2)}\frac{\mathrm{d}j_l^{\mathrm{i}}}{\mathrm{d}r} - j_l^{\mathrm{i}}\frac{\mathrm{d}h_l^{(2)}}{\mathrm{d}r}\right)\left(h_l^{(1)}\frac{\mathrm{d}(rj_l^{\mathrm{i}})}{r\mathrm{d}r} - n^2 j_l^{\mathrm{i}}\frac{\mathrm{d}(rh_l^{(1)})}{r\mathrm{d}r}\right) - \frac{\alpha^2\Theta^2}{\pi^2}h_l^{(2)}\frac{\mathrm{d}(rj_l^{\mathrm{i}})}{r\mathrm{d}r}j_l^{\mathrm{i}}\frac{\mathrm{d}(rh_l^{(1)})}{r\mathrm{d}r}}{\left(h_l^{(1)}\frac{\mathrm{d}j_l^{\mathrm{i}}}{\mathrm{d}r} - j_l^{\mathrm{i}}\frac{\mathrm{d}h_l^{(1)}}{\mathrm{d}r}\right)\left(h_l^{(1)}\frac{\mathrm{d}(rj_l^{\mathrm{i}})}{r\mathrm{d}r} - n^2 j_l^{\mathrm{i}}\frac{\mathrm{d}(rh_l^{(1)})}{r\mathrm{d}r}\right) - \frac{\alpha^2\Theta^2}{\pi^2}h_l^{(1)}\frac{\mathrm{d}(rj_l^{\mathrm{i}})}{r\mathrm{d}r}j_l^{\mathrm{i}}\frac{\mathrm{d}(rh_l^{(1)})}{r\mathrm{d}r}}e_l^{(2)}. \quad (33)
\end{aligned}
$$

Noting that we are ultimately interested in the computation of the induced outgoing wave (i.e., total outgoing wave subtracted by outgoing Hankel functions contained in the external plane wave), we finally define a 2x2 scattering matrix $\boldsymbol{t}_l$ by $\boldsymbol{d}_l - \boldsymbol{e}_l = \boldsymbol{t}_l \boldsymbol{e}_l$. Terms of the kind $x p_l(x)$ can be identified as spherical Riccati–Bessel functions and will be denoted by $\tilde{p}_l(x) = x p_l(x)$ from now on (here $p_l$ represents $j_l^{\mathrm{i}}, j_l^{\mathrm{o}}, y_l, h_l^1$ and $h_l^2$). For clarity we furthermore define the arguments of the Bessel and Hankel functions as $\rho^{\mathrm{i}} = nkr$ and $\rho^{\mathrm{o}} = kr$ for the inner and outer region of the sphere respectively. The components of the scattering matrix $\boldsymbol{t}_l$ than read:

$$t_l^{11} = \frac{\left(h_l^{(1)} \frac{\mathrm{d}\tilde{j}_l^{\mathrm{i}}}{\mathrm{d}\rho^{\mathrm{i}}} - j_l^{\mathrm{i}} \frac{\mathrm{d}\tilde{h}_l^{(1)}}{\mathrm{d}\rho^{\mathrm{o}}}\right)\left(j_l^{\mathrm{o}} \frac{\mathrm{d}\tilde{j}_l^{\mathrm{i}}}{\mathrm{d}\rho^{\mathrm{i}}} - n^2 j_l^{\mathrm{i}} \frac{\mathrm{d}\tilde{j}_l^{\mathrm{o}}}{\mathrm{d}\rho^{\mathrm{o}}}\right) - \frac{\alpha^2 \Theta^2}{\pi^2} h_l^{(1)} \frac{\mathrm{d}\tilde{j}_l^{\mathrm{i}}}{\mathrm{d}\rho^{\mathrm{i}}} j_l^{\mathrm{i}} \frac{\mathrm{d}\tilde{j}_l^{\mathrm{o}}}{\mathrm{d}\rho^{\mathrm{o}}}}{\left(h_l^{(1)} \frac{\mathrm{d}\tilde{j}_l^{\mathrm{i}}}{\mathrm{d}\rho^{\mathrm{i}}} - j_l^{\mathrm{i}} \frac{\mathrm{d}\tilde{h}_l^{(1)}}{\mathrm{d}\rho^{\mathrm{o}}}\right)\left(h_l^{(1)} \frac{\mathrm{d}\tilde{j}_l^{\mathrm{i}}}{\mathrm{d}\rho^{\mathrm{i}}} - n^2 j_l^{\mathrm{i}} \frac{\mathrm{d}\tilde{h}_l^{(1)}}{\mathrm{d}\rho^{\mathrm{o}}}\right) - \frac{\alpha^2 \Theta^2}{\pi^2} h_l^{(1)} \frac{\mathrm{d}\tilde{j}_l^{\mathrm{i}}}{\mathrm{d}\rho^{\mathrm{i}}} j_l^{\mathrm{i}} \frac{\mathrm{d}\tilde{h}_l^{(1)}}{\mathrm{d}\rho^{\mathrm{o}}}} , \tag{34}$$

$$t_l^{12} = -\frac{\frac{\alpha l(l+1)\Theta}{kR\pi} j_l^{\mathrm{i}} \frac{\mathrm{d}\tilde{j}_l^{\mathrm{i}}}{\mathrm{d}\rho^{\mathrm{i}}}\left(\frac{\mathrm{d}\tilde{y}_l}{\mathrm{d}\rho^{\mathrm{o}}} j_l^{\mathrm{o}} - \frac{\mathrm{d}\tilde{j}_l^{\mathrm{o}}}{\mathrm{d}\rho^{\mathrm{o}}} y_l\right)}{\left(h_l^{(1)} \frac{\mathrm{d}\tilde{j}_l^{\mathrm{i}}}{\mathrm{d}\rho^{\mathrm{i}}} - j_l^{\mathrm{i}} \frac{\mathrm{d}\tilde{h}_l^{(1)}}{\mathrm{d}\rho^{\mathrm{o}}}\right)\left(h_l^{(1)} \frac{\mathrm{d}\tilde{j}_l^{\mathrm{i}}}{\mathrm{d}\rho^{\mathrm{i}}} - n^2 j_l^{\mathrm{i}} \frac{\mathrm{d}\tilde{h}_l^{(1)}}{\mathrm{d}\rho^{\mathrm{o}}}\right) - \frac{\alpha^2 \Theta^2}{\pi^2} h_l^{(1)} \frac{\mathrm{d}\tilde{j}_l^{\mathrm{i}}}{\mathrm{d}\rho^{\mathrm{i}}} j_l^{\mathrm{i}} \frac{\mathrm{d}\tilde{h}_l^{(1)}}{\mathrm{d}\rho^{\mathrm{o}}}} , \tag{35}$$

$$t_l^{21} = \frac{\frac{\alpha \Theta kR}{\pi l(l+1)} j_l^{\mathrm{i}} \frac{\mathrm{d}\tilde{j}_l^{\mathrm{i}}}{\mathrm{d}\rho^{\mathrm{i}}}\left(j_l^{\mathrm{o}} \frac{\mathrm{d}\tilde{y}_l}{\mathrm{d}\rho^{\mathrm{o}}} - y_l \frac{\mathrm{d}\tilde{j}_l^{\mathrm{o}}}{\mathrm{d}\rho^{\mathrm{o}}}\right)}{\left(h_l^{(1)} \frac{\mathrm{d}\tilde{j}_l^{\mathrm{i}}}{\mathrm{d}\rho^{\mathrm{i}}} - j_l^{\mathrm{i}} \frac{\mathrm{d}\tilde{h}_l^{(1)}}{\mathrm{d}\rho^{\mathrm{o}}}\right)\left(h_l^{(1)} \frac{\mathrm{d}\tilde{j}_l^{\mathrm{i}}}{\mathrm{d}\rho^{\mathrm{i}}} - n^2 j_l^{\mathrm{i}} \frac{\mathrm{d}\tilde{h}_l^{(1)}}{\mathrm{d}\rho^{\mathrm{o}}}\right) - \frac{\alpha^2 \Theta^2}{\pi^2} h_l^{(1)} \frac{\mathrm{d}\tilde{j}_l^{\mathrm{i}}}{\mathrm{d}\rho^{\mathrm{i}}} j_l^{\mathrm{i}} \frac{\mathrm{d}\tilde{h}_l^{(1)}}{\mathrm{d}\rho^{\mathrm{o}}}} , \tag{36}$$

$$t_l^{22} = \frac{\left(j_l^{\mathrm{o}} \frac{\mathrm{d}\tilde{j}_l^{\mathrm{i}}}{\mathrm{d}\rho^{\mathrm{i}}} - j_l^{\mathrm{i}} \frac{\mathrm{d}\tilde{j}_l^{\mathrm{o}}}{\mathrm{d}\rho^{\mathrm{o}}}\right)\left(h_l^{(1)} \frac{\mathrm{d}\tilde{j}_l^{\mathrm{i}}}{\mathrm{d}\rho^{\mathrm{i}}} - n^2 j_l^{\mathrm{i}} \frac{\mathrm{d}\tilde{h}_l^{(1)}}{\mathrm{d}\rho^{\mathrm{o}}}\right) - \frac{\alpha^2 \Theta^2}{\pi^2} j_l^{\mathrm{o}} \frac{\mathrm{d}\tilde{j}_l^{\mathrm{i}}}{\mathrm{d}\rho^{\mathrm{i}}} j_l^{\mathrm{i}} \frac{\mathrm{d}\tilde{h}_l^{(1)}}{\mathrm{d}\rho^{\mathrm{o}}}}{\left(h_l^{(1)} \frac{\mathrm{d}\tilde{j}_l^{\mathrm{i}}}{\mathrm{d}\rho^{\mathrm{i}}} - j_l^{\mathrm{i}} \frac{\mathrm{d}\tilde{h}_l^{(1)}}{\mathrm{d}\rho^{\mathrm{o}}}\right)\left(h_l^{(1)} \frac{\mathrm{d}\tilde{j}_l^{\mathrm{i}}}{\mathrm{d}\rho^{\mathrm{i}}} - n^2 j_l^{\mathrm{i}} \frac{\mathrm{d}\tilde{h}_l^{(1)}}{\mathrm{d}\rho^{\mathrm{o}}}\right) - \frac{\alpha^2 \Theta^2}{\pi^2} h_l^{(1)} \frac{\mathrm{d}\tilde{j}_l^{\mathrm{i}}}{\mathrm{d}\rho^{\mathrm{i}}} j_l^{\mathrm{i}} \frac{\mathrm{d}\tilde{h}_l^{(1)}}{\mathrm{d}\rho^{\mathrm{o}}}} , \tag{37}$$

respectively. Accordingly, the classical Mie theory result for the scattering matrix, which only contains diagonal entries (see, e.g., [27]), is recovered when dropping the topological term. Note furthermore that we get an additional factor of $-\frac{1}{2}$ in the definition of $\boldsymbol{t}_l$ in comparison to the well-known classical result, which is a consequence of our choice of the vector spherical harmonics defined by Barrera et al. [31], deviating from those used conventionally. Note, that we already correct for the factor $-\frac{1}{2}$ in the Eqs. (34) - (37).

The scattering cross section $\sigma_{\mathrm{s}}$ and extinction cross section $\sigma_{\mathrm{e}}$ then reads:

$$\sigma_{\mathrm{s}} = \frac{2\pi}{k^2} \sum_{l=0}^{\infty} (2l+1)\left(\left(t_l^{11} + t_l^{12}\right)^2 + \left(t_l^{21} + t_l^{22}\right)^2\right) \tag{38}$$

and

$$\sigma_{\mathrm{e}} = \frac{2\pi}{k^2} \sum_{l=0}^{\infty} (2l+1)\,\Re\left\{t_l^{11} + t_l^{12} + t_l^{21} + t_l^{22}\right\} \tag{39}$$

(see Eq. (81) and Eq. (83) for the full analytic solution). Accordingly, we observe that the well-known classical Mie theory result ($\Theta = 0$, see, e.g., [27]) is modified by a topological term of the order $\alpha^2$ in both the nominator and denominator. While the former slightly shifts the energy of the resonances at zeros of the nominator the latter leads to a modification of the scattering intensity. Due to their smallness both are beyond current experimental detection.

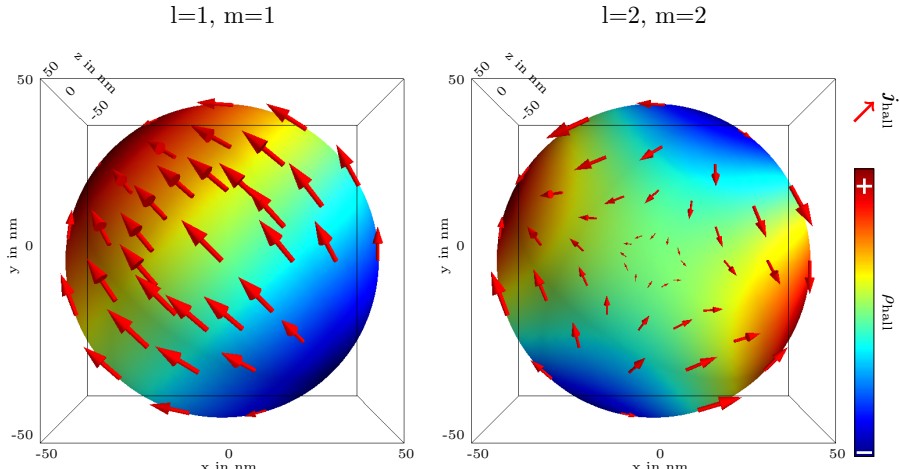

Figure 3: Hall currents (real part) of the electric $l = 1, m = 1 / l = 2, m = 2$ mode (vector plot) and the corresponding oscillating charges (color coded) for a TI sphere ($R = 50$ nm, $\epsilon = -1$, $\Theta = \pi$) embedded in vacuum at $\hbar\omega = 2$ eV. Note that the induced hall charges and currents are coupled by a continuity equation. The plotted quantities stem from the axion related terms of the boundary conditions (Eq. (23) and Eq. (24)) vanishing for topological trivial materials.

## Inhomogeneous Case

We finally turn our attention to the general inhomogeneous case. We first note that the solution of the pertaining differential Eqs. (18) and (20) for arbitrary external currents and charges generally does not admit analytical solutions and ultimately requires numerical methods. These consist of (A) expanding the external charge gradients and current sources into vector spherical harmonics, (B) imposing suitable boundary conditions (e.g., Dirichlet boundary conditions at $r = 0$ and $r \to \infty$), (C) converting the ordinary differential equations into an algebraic system of equations by imposing a suitable grid, and (D) inverting this system to obtain a numerical solution for each $E_{lm}$. Highly symmetric charge and current distributions (such as the homogeneously charged sphere) may be treated analytically.

Having in mind an experimental setups facilitating a direct measurement of the axion response, we focus on the dielectric response to an external charge moving along a straight line in the following. This setup corresponds to probing the dielectric response of nanostructures by measuring the energy loss and deflection of a highly-focused and highly-accelerated electron beam in the Transmission Electron Microscope (TEM). This Electron Energy Loss Spectroscopy (EELS) technique is widely employed for studying dielectric phenomena such as surface plasmons at the nanoscale and we will come up with a novel detection setup suited to analyze the axion term as a result of the following considerations.

The charge density and current (in frequency space) of an electron moving with velocity $v$ in $z$-direction read

$$\rho(\boldsymbol{r}, \omega) = -\frac{e}{v}\delta(\boldsymbol{r}_\perp - \boldsymbol{r}_{0\perp})e^{i\frac{\omega}{v}(z-z_0)} \tag{40}$$

and

$$\boldsymbol{j}(\boldsymbol{r}, \omega) = -e\delta(\boldsymbol{r}_\perp - \boldsymbol{r}_{0\perp})e^{i\frac{\omega}{v}(z-z_0)}\hat{\boldsymbol{z}} \tag{41}$$

respectively. Here, $\boldsymbol{r}_{0\perp}$ denotes the impact parameter in the $x - y$ plane and $\hat{\boldsymbol{z}}$ is the unit vector in $z$-direction. It is obvious that such an external source defies any spherical symmetry, hence would require the tedious numerical treatment as outlined above. In the following

we will therefore adopt several approximations as described in Ref. [35], ultimately leading to analytical expressions for the energy loss and the deflection of the electron beam. These approximations consist of neglecting (A) possible longitudinal solutions (only the projection on the transverse components is considered) and (B) certain boundary terms in the poloidal-toroidal expansion of the transverse solutions [35], which, however, are typically justified for energy losses of fast electrons in the optical regime. The benefit of these restrictions is that one can directly adopt the results of the homogeneous calculations treating the electron as a source of external poloidal and toroidal fields instead as a moving external charge with $\rho \neq 0$. The energy loss $\Delta E$ suffered by a fast electron is determined by the path integral of the force along the beam path $z$

$$
\begin{aligned}
\Delta E &= \int dt\, \boldsymbol{v} \cdot \boldsymbol{E}^{\mathrm{ind}}(z(t), t) \\
&= \frac{1}{\pi} \int_0^\infty d\omega \int_{-\infty}^\infty dz\, \Re\left\{ e^{-i\frac{\omega}{v}z} \tilde{E}_z^{\mathrm{ind}}(\boldsymbol{r}_{0\perp}, z, \omega) \right\} \\
&= \int_0^\infty \Gamma^{\mathrm{loss}}(\omega)\, \omega\, d\omega,
\end{aligned}
\tag{42}
$$

where we introduced the loss probability $\Gamma^{\mathrm{loss}}(\omega)$ corresponding to the EEL spectrum. Repeating the derivation detailed in Ref. [35] and introducing the axion term the latter reads

$$
\Gamma^{\mathrm{loss}}(\omega) = \frac{1}{c\omega} \sum_{l,m} K_m^2\left(\frac{\omega b}{v\gamma}\right) \left( C_{lm}^\perp \Im\left\{ t_l^{11} + t_l^{21} \right\} + C_{lm}^{(2)} \Im\left\{ t_l^{12} + t_l^{22} \right\} \right).
\tag{43}
$$

Here, $K_m$ denotes the modified Bessel function of order $m$ and $\gamma$ is the Lorentz factor. The positive coefficients

$$
C_{lm}^\perp = \frac{1}{l(l+1)} \left| \frac{2mv}{c} A_{lm} \right|^2
\tag{44}
$$

and

$$
C_{lm}^{(2)} = \frac{1}{l(l+1)} \left| \frac{1}{\gamma} B_{lm} \right|^2
\tag{45}
$$

are functions of $v/c$ only as detailed in Appendix A.3 (the definition of $A_{lm}$ and $B_{lm}$ can be found in Eq. (98) and Eq. (92) respectively). Inspecting the above expressions for the loss probability (Eq. (43)) we observe that the peaks in the loss probabilities, which correspond to zeros in the denominators of the scattering matrix elements, are shifted by terms of $\mathcal{O}(\alpha^2)$ due to the axion terms in complete analogy to the photon scattering case. Consequently, directly probing the axion term by measuring peak positions in the loss probability is out of range with current TEM-EELS instrumentation. There is, however, a contribution linear in $\alpha$ to the magnitude of the loss probability, which may be detectable. Note, however, that a measurement of such small modifications on top of the total loss probability requires precise reference measurements on identical non-topological samples, which is typically not possible. Similar to the photon scattering case a direct measurement of the off-diagonal terms of the scattering matrix could solve that problem. To that end we consider the lateral deflection of the electron beam given by

$$
\begin{pmatrix} p_x^{\mathrm{kin}} \\ p_y^{\mathrm{kin}} \end{pmatrix} = -\frac{1}{\pi v} \int_0^\infty d\omega \int dz\, \Re\left\{ e^{-i\frac{\omega}{v}z} \left( \begin{pmatrix} E_x^{\mathrm{ind}}(z, \omega) \\ E_y^{\mathrm{ind}}(z, \omega) \end{pmatrix} + \frac{v}{c} \begin{pmatrix} -B_y^{\mathrm{ind}}(z, \omega) \\ B_x^{\mathrm{ind}}(z, \omega) \end{pmatrix} \right) \right\}.
\tag{46}
$$

We now restrict ourselves to a discussion of the lateral deflection component being tangential to the surface of the sphere (see Fig. 5). The logic behind is that such a deflection is not present

in the non-topological case. It is generated by a toroidal electric and a poloidal magnetic field, which are both induced by the off-diagonal $t^{21}$ coupling term from the dominant poloidal electric field. The static analogue to this field, referred to as a Pearl vortex, has been discussed at length in Ref. [22].

Seeking the computation of the axion-induced toroidal (poloidal) electric (magnetic) field, we only consider $d_l^{(2)} = t_l^{21} e_l^\perp$ in the following. Performing similar manipulation than those leading to Eq. (43) the transferred lateral momentum may be expressed in terms of scattering matrix elements (see Appendix A.3 for details)

$$p_y^{\text{kin}}(b,0) = \frac{2}{c^2\gamma} \int_0^\infty d\omega \sum_{l,m} K_m \frac{B_{lm}}{l(l+1)} \left( \Re\left\{ D_{lm}^+ t_{l21} \right\} - \Re\left\{ D_{lm}^- t_{l21} \right\} \right), \qquad (47)$$

where

$$D_{lm}^+ = \sqrt{(l+m)(l-m+1)} A_{lm-1}^* K_{m-1}\left( \frac{\omega b}{v\gamma} \right), \qquad (48)$$

$$D_{lm}^- = \sqrt{(l-m)(l+m+1)} A_{lm+1}^* K_{m+1}\left( \frac{\omega b}{v\gamma} \right). \qquad (49)$$

We finally calculate the transverse momentum per electron at a certain energy loss $\omega$ (experimentally energy loss interval around $\omega$) as recorded by EELS. To that end we normalize the spectral representation of the transverse momentum by the number of electrons being inelastically scattered at a particular energy loss.

$$p_y^{\text{kin}}(\omega) = \frac{2}{c^2\gamma} \sum_{l,m} K_m \frac{B_{lm}}{l(l+1)} \frac{\left( \Re\left\{ D_{lm}^+ t_{l21} \right\} - \Re\left\{ D_{lm}^- t_{l21} \right\} \right)}{\Gamma^{\text{loss}}(\omega)}. \qquad (50)$$

At this point we finally need computational support in order to calculate the loss probability and the transverse momentum transfer. Fortunately the classical case ($\Theta = 0$) is already implemented within the MNPBEM toolbox of U. Hohenester [36]. However, we need to add the computation of the off-diagonal elements of the scattering matrix ($t_l^{12}$ and $t_l^{21}$) vanishing in the topological trivial case as well as the axion contribution of the diagonal elements ($t_l^{11}$ and $t_l^{22}$). After this generalization of the existing implementation we can calculate the loss probability and the lateral deflection for a real scenario. For an electron with 40 keV primary energy a comparison between a topological trivial and a TI sphere is shown in Fig. 4 (loss probability on the left hand side and lateral deflection angle on the right hand side). Here the smallness of the axion contribution to the loss probability manifests itself in almost identical curves for the classical ($\Theta = 0$) and the topological ($\Theta = \pi$) case. The angle of lateral deflection $\beta$ vanishing in the topological trivial case was calculated to a maximum value of $\approx 4.5 \times 10^{-2} \mu$rad for the TI sphere. This angle is at the edge of current detection limits using advanced TEM setups like inelastic momentum transfer (IMT) (see Ref. [37]). This method detects the deflection angle of fast electrons due to induced electromagnetic fields excited by the same electron (see Fig. 5). The calculated value of $\beta$ is roughly one order of magnitude smaller than the angular resolution achievable with the IMT setup applied in Ref. [37]. Note, however, that the lateral deflection depends (in strictly difference to the value of $\Theta$) strongly on the geometry of the sample and is potentially larger for non-spherical particles, e.g., cuboids exhibiting long straight faces. To calculate the response of arbitrary shaped particles numerical solvers are indispensable. Furthermore the experimental IMT setup can be further improved in terms of increased angular resolution.

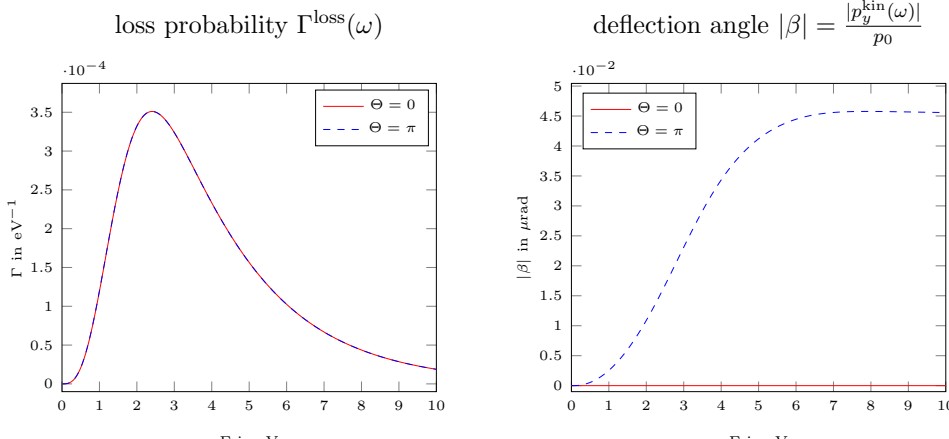

Figure 4: Loss probability $\Gamma$ (left hand side) and absolute value of the deflection angle $|\beta| = |p_y^{\text{kin}}|/p_0$ (right hand side) of the exciting electron for a topological trivial (red solid curves respectively) and TI (blue dashed curves respectively) sphere with $R = 50\,\text{nm}$ and $\epsilon = -0.5$. The impact parameter of the exciting electron was set to $1\,\text{nm}$ at $40\,\text{keV}$ primary energy. Note that the loss probability follows almost the same course for the topological trivial and the TI case due to the tiny contribution of the axion term to the loss probability quadratic in $\alpha$.

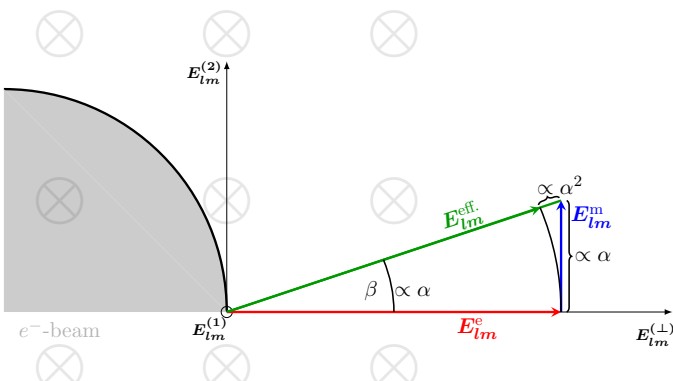

Figure 5: Axion admixing of magnetic $\left(E_{lm}^{\text{m}}\right)$ components to an electric $\left(E_{lm}^{\text{e}}\right)$ mode. The change of the lengths of the resulting effective electric field vector $E_{lm}^{\text{eff.}}$ is of the order $\alpha^2$ whereas the lateral deflection angle $\beta$ scales linear with $\alpha$.

## 3 Conclusion

In summary, we have extended the description of the electromagnetic response of spherical particles to include the ramifications of an axion term such as present in topological insulators. We have studied the consequences for the excitation of localized surface plasmons on spherical TI nanoparticles. We found an intrinsic mixing of electric and magnetic modes as leading order effect (linear in $\alpha$). This mixing manifests itself as a small rotation of the induced electric and magnetic fields, which we anticipate to be detectable by suitable setups. Promising candidates utilizing localized probes (i.e. electron beams) are polarization-resolved cathodoluminescence and inelastic momentum transfer measurements [37]. Corresponding shifts in LSP resonance

energies and scattering or extinction coefficients, on the other hand, are of the order of $\alpha^2$, hence difficult to detect. We also note that the presented formalism allows to include arbitrary external currents and charges, typically not included in the standard formulation of Mie theory. As an example we discussed the dielectric response of a TI sphere transmitted by a focused electron beam as utilized in electron energy loss spectroscopy.

# Acknowledgments

A.L. received funding from the European Research Council (ERC) under the Horizon 2020 research and innovation program of the European Union (grant agreement no. 715620). J.S. received funding from the Deutsche Forschungsgemeinschaft (DFG, German Research Foundation) under Germany's Excellence Strategy through Würzburg-Dresden Cluster of Excellence on Complexity and Topology in Quantum Matter - ct.qmat (EXC 2147, project-id 390858490). JvdB acknowledges support from the Deutsche Forschungsgemeinschaft (DFG) through the Collaborative Research Center (Sonderforschungsbereich) SFB 1143 (Project No. 247310070).

# A  Mie Theory

## A.1  Field equations in spherical coordinates

The general dielectric response of the sphere (i.e. Mie theory) is obtained by solving the response Eq. (15) exploiting spherical symmetry. In the following we tackle the problem by a piecewise solution in regions of constant $\epsilon$ and $\Theta$ (i.e., within and outside of the sphere) and fixing the missing integration constants through appropriate boundary conditions. Accordingly, the response equation in homogeneous regions reads

$$\nabla \times \nabla \times \boldsymbol{E} - k^2 \epsilon \boldsymbol{E} = -i\frac{4\pi k}{c}\boldsymbol{j} \tag{51}$$

or

$$\triangle \boldsymbol{E} + k^2 \epsilon \boldsymbol{E} = i\frac{4\pi k}{c}\boldsymbol{j} + \frac{4\pi}{\epsilon}\boldsymbol{\nabla}\rho \,. \tag{52}$$

In the next step we expand the solution into vector spherical harmonics (following the definition of Barrera et al. [31])

$$\mathbf{E}(\boldsymbol{r}) \;=\; \sum_{l=0}^{\infty}\sum_{m=-l}^{l}\left(E_{lm}^{\perp}(r)\mathbf{Y}_{lm} + E_{lm}^{(1)}(r)\boldsymbol{\Psi}_{lm} + E_{lm}^{(2)}(r)\boldsymbol{\Phi}_{lm}\right), \tag{53}$$

which gives us later less headaches, when defining the boundary conditions, compared to the more common Hertz potential expansion. Inserting the expansion into Eq. (51) yields

$$\boldsymbol{\nabla} \times \sum_{l=0}^{\infty}\sum_{m=-l}^{l}\left[-\frac{l(l+1)}{r}E_{lm}^{(2)}\mathbf{Y}_{lm} - \left(\frac{\mathrm{d}E_{lm}^{(2)}}{\mathrm{d}r} + \frac{1}{r}E_{lm}^{(2)}\right)\boldsymbol{\Psi}_{lm}\right.$$

$$\left. + \left(-\frac{1}{r}E_{lm}^{\perp} + \frac{\mathrm{d}E_{lm}^{(1)}}{\mathrm{d}r} + \frac{1}{r}E_{lm}^{(1)}\right)\boldsymbol{\Phi}_{lm}\right] - k^2\epsilon\boldsymbol{E} = -i\frac{4\pi k}{c}\boldsymbol{j} \tag{54}$$

after evaluating one curl and finally

$$\sum_{l=0}^{\infty}\sum_{m=-l}^{l}\left[\left(\frac{l(l+1)}{r^2}E_{lm}^{\perp}-\frac{l(l+1)\mathrm{d}E_{lm}^{(1)}}{r\mathrm{d}r}-\frac{l(l+1)}{r^2}E_{lm}^{(1)}-k^2\epsilon E_{lm}^{\perp}\right)\mathbf{Y}_{lm}\right.$$
$$+\left(\frac{1}{r}\frac{\mathrm{d}E_{lm}^{\perp}}{\mathrm{d}r}-\frac{\mathrm{d}^2E_{lm}^{(1)}}{\mathrm{d}r^2}-\frac{2}{r}\frac{\mathrm{d}E_{lm}^{(1)}}{\mathrm{d}r}-k^2\epsilon E_{lm}^{(1)}\right)\boldsymbol{\Psi}_{lm}$$
$$\left.+\left(\frac{l(l+1)}{r^2}E_{lm}^{(2)}-\frac{\mathrm{d}^2E_{lm}^{(2)}}{\mathrm{d}r^2}-\frac{2}{r}\frac{\mathrm{d}E_{lm}^{(2)}}{\mathrm{d}r}-k^2\epsilon E_{lm}^{(2)}\right)\boldsymbol{\Phi}_{lm}\right]=-i\frac{4\pi k}{c}\boldsymbol{j} \tag{55}$$

or, upon expanding Eq. (52),

$$\sum_{l=0}^{\infty}\sum_{m=-l}^{l}\left[\left(\frac{2+l(l+1)}{r^2}E_{lm}^{\perp}-\frac{2}{r}\frac{\mathrm{d}E_{lm}^{\perp}}{\mathrm{d}r}-\frac{\mathrm{d}^2E_{lm}^{\perp}}{\mathrm{d}r^2}-\frac{2l(l+1)}{r^2}E_{lm}^{(1)}-k^2\epsilon E_{lm}^{\perp}\right)\mathbf{Y}_{lm}\right.$$
$$+\left(-\frac{2}{r^2}E_{lm}^{\perp}-\frac{2}{r}\frac{\mathrm{d}E_{lm}^{(1)}}{\mathrm{d}r}-\frac{\mathrm{d}^2E_{lm}^{(1)}}{\mathrm{d}r^2}+\frac{l(l+1)}{r^2}E_{lm}^{(1)}-k^2\epsilon E_{lm}^{(1)}\right)\boldsymbol{\Psi}_{lm}$$
$$\left.+\left(-\frac{2}{r}\frac{\mathrm{d}E_{lm}^{(2)}}{\mathrm{d}r}-\frac{\mathrm{d}^2E_{lm}^{(2)}}{\mathrm{d}r^2}+\frac{l(l+1)}{r^2}E_{lm}^{(2)}-k^2\epsilon E_{lm}^{(2)}\right)\boldsymbol{\Phi}_{lm}\right]=-i\frac{4\pi k}{c}\boldsymbol{j}-\frac{4\pi}{\epsilon}\boldsymbol{\nabla}\rho\,. \tag{56}$$

For notional simplicity we did not write the vector harmonics expansion on the RHS (inhomogeneous term) of both equations, which is straightforward, however. Taking into account the closure relations of the vector spherical harmonics both equations reduce to a system of three ordinary differential equations for each vector spherical harmonics. Note furthermore that only the first two are coupled, respectively, whereas the last one can be solved independently. We proceed by first solving the coupled system of the first two. Subtracting the $\boldsymbol{\Psi}_{lm}$ expansion coefficients of both equations gives

$$\frac{1}{r}\frac{\mathrm{d}E_{lm}^{\perp}}{\mathrm{d}r}+\frac{2}{r^2}E_{lm}^{\perp}-\frac{l(l+1)}{r^2}E_{lm}^{(1)}=\frac{4\pi}{\epsilon}(\boldsymbol{\nabla}\rho)_{lm}^{(1)} \tag{57}$$

or

$$E_{lm}^{(1)}=\frac{r}{l(l+1)}\frac{\mathrm{d}E_{lm}^{\perp}}{\mathrm{d}r}+\frac{2}{l(l+1)}E_{lm}^{\perp}-\frac{4\pi}{\epsilon l(l+1)}r^2(\boldsymbol{\nabla}\rho)_{lm}^{(1)}\,. \tag{58}$$

Inserting into the $\mathbf{Y}_{lm}$ expansion coefficient of Eq. (55) then leads to

$$\frac{l(l+1)}{r^2}E_{lm}^{\perp}-\frac{l(l+1)}{r}\frac{\mathrm{d}E_{lm}^{(1)}}{\mathrm{d}r}-\frac{1}{r}\frac{\mathrm{d}E_{lm}^{\perp}}{\mathrm{d}r}-\frac{2}{r^2}E_{lm}^{\perp}-k^2\epsilon E_{lm}^{\perp}=-i\frac{4\pi k}{c}j_{lm}^{\perp}-\frac{4\pi}{\epsilon}(\boldsymbol{\nabla}\rho)_{lm}^{(1)}\,. \tag{59}$$

We finally compute the derivative of the $E_{lm}^{(1)}$ term by inserting Eq. (58)

$$-\frac{1}{r}\frac{\mathrm{d}}{\mathrm{d}r}\left(\frac{r\mathrm{d}E_{lm}^{\perp}}{\mathrm{d}r}+2E_{lm}^{\perp}-\frac{4\pi}{\epsilon}r^2(\boldsymbol{\nabla}\rho)_{lm}^{(1)}\right)-\frac{1}{r}\frac{\mathrm{d}E_{lm}^{\perp}}{\mathrm{d}r}+\left(\frac{l(l+1)-2}{r^2}-k^2\epsilon\right)E_{lm}^{\perp}$$
$$=-i\frac{4\pi k}{c}j_{lm}^{\perp}-\frac{4\pi}{\epsilon}(\boldsymbol{\nabla}\rho)_{lm}^{(1)} \tag{60}$$

obtaining

$$r^2\frac{\mathrm{d}^2E_{lm}^{\perp}}{\mathrm{d}r^2}+4r\frac{\mathrm{d}E_{lm}^{\perp}}{\mathrm{d}r}+\left(k^2\epsilon r^2-l(l+1)+2\right)E_{lm}^{\perp}=i\frac{4\pi kr^2}{c}j_{lm}^{\perp}+\frac{12\pi r^2}{\epsilon}(\boldsymbol{\nabla}\rho)_{lm}^{(1)}+\frac{4\pi r^3}{\epsilon}\frac{\mathrm{d}(\boldsymbol{\nabla}\rho)_{lm}^{(1)}}{\mathrm{d}r} \tag{61}$$

or (after setting $E_{lm}^{\perp} = E_{lm}^{(\perp)}/r$)

$$r^2 \frac{\mathrm{d}^2 E_{lm}^{(\perp)}}{\mathrm{d}r^2} + 2r \frac{\mathrm{d} E_{lm}^{(\perp)}}{\mathrm{d}r} + \left(k^2 \epsilon r^2 - l(l+1)\right) E_{lm}^{(\perp)} = -i \frac{4\pi k}{c} r^3 j_{lm}^{\perp} + \frac{12\pi}{\epsilon} r^3 (\boldsymbol{\nabla}\rho)_{lm}^{(1)} + \frac{4\pi}{\epsilon} r^4 \frac{\mathrm{d}(\boldsymbol{\nabla}\rho)_{lm}^{(1)}}{\mathrm{d}r}. \tag{62}$$

This is an inhomogeneous second order differential equation in $E_{lm}^{(\perp)}$ of (modified) spherical Bessel equation type (after absorbing $k\sqrt{|\epsilon|} = kn$ into $r$). The third equation has the same structure

$$r^2 \frac{\mathrm{d}^2 E_{lm}^{(2)}}{\mathrm{d}r^2} + 2r \frac{\mathrm{d} E_{lm}^{(2)}}{\mathrm{d}r} + \left(k^2 \epsilon r^2 - l(l+1)\right) E_{lm}^{(2)} = i \frac{4\pi k r^2}{c} j_{lm}^{(2)}. \tag{63}$$

Both sets of solutions are independent. Note furthermore that the solution space can be further restricted by only admitting solutions, which do not diverge at the origin (i.e. modified spherical Bessel functions of the first kind in the interior). The homogeneous solution to the first expansion then explicitly reads

$$E_{lm}^{(\perp)}(r) = \begin{cases} a_l^{(\perp)} j_l^{\mathrm{i}}(nkr), & r < R \\ b_l^{(\perp)} j_l^{\mathrm{o}}(kr) + c_l^{(\perp)} y_l(kr), & r \geq R, \end{cases} \tag{64}$$

and from Eq. (58)

$$E_{lm}^{(1)}(r) = \begin{cases} a_l^{(\perp)} \left[ \frac{nk}{l(l+1)} j_{l+1}^{\mathrm{i}}(nkr) + \frac{1}{l} \frac{j_l^{\mathrm{i}}(nkr)}{r} \right], & r < R \\ b_l^{(\perp)} \frac{rkj_{l-1}^{\mathrm{o}}(kr) + lj_l^{\mathrm{o}}(kr)}{rl(l+1)} + c_l^{(\perp)} \frac{rky_{l-1}(kr) + ly_l(kr)}{rl(l+1)}, & r \geq R. \end{cases} \tag{65}$$

The second solution reads

$$E_{lm}^{(2)}(r) = \begin{cases} a_l^{(2)} j_l^{\mathrm{i}}(nkr), & r < R \\ b_l^{(2)} j_l^{\mathrm{o}}(kr) + c_l^{(2)} y_l(kr), & r \geq R. \end{cases} \tag{66}$$

In the first case the magnetic field is strictly tangential to the sphere, whereas in the second case the electric field is tangential. In accordance with the convention employed for the plane boundary (i.e. sphere radius $\rightarrow \infty$) they are called magnetic (m) and electric (e) modes. Note furthermore that in both cases the homogeneous solutions are degenerate with respect to $m$.

## A.2 Boundary Conditions

In order to determine the 6 expansion coefficients of the homogeneous solutions above we require 4 BCs (two for $E^{(\perp)}$ and two for $E^{(2)}$) for each homogeneous domain, $E^{(1)}$ can be computed once $E^{(\perp)}$ is known. The two remaining degrees of freedom are fixed by normalization of the electric and magnetic modes respectively. We begin with the internal boundaries at the surface of the sphere. Indeed, it is only at this stage, where deviations from the classical Mie theory are introduced by the axion term as we anticipated. These boundary conditions are derived from partial derivatives normal to the surface as usual:

1. Boundary condition from the second Maxwell equation

$$E_{lm1}^{(1)} = E_{lm2}^{(1)} \tag{67}$$

   and hence

$$\frac{r\mathrm{d}E_{lm1}^{\perp}}{\mathrm{d}r} + 2E_{lm1}^{\perp} = \frac{r\mathrm{d}E_{lm2}^{\perp}}{\mathrm{d}r} + 2E_{lm2}^{\perp}. \tag{68}$$

2. Boundary condition from the third Maxwell equation

$$\hat{r} \cdot \boldsymbol{B}_2 - \hat{r} \cdot \boldsymbol{B}_1 = 0, \tag{69}$$

$$-\frac{l(l+1)}{r}\left(E^{(2)}_{lm1} - E^{(2)}_{lm2}\right) = 0. \tag{70}$$

3. Boundary condition from the first Maxwell equation ($\hat{r} \,\widehat{=}\,$ outward unit normal)

$$\hat{r}\boldsymbol{D}_2 - \hat{r}\boldsymbol{D}_1 = \frac{i\alpha\Theta_2}{k\pi}\hat{r}\cdot(\nabla\times\boldsymbol{E}_2) - \frac{i\alpha\Theta_1}{k\pi}\hat{r}\cdot(\nabla\times\boldsymbol{E}_1) \tag{71}$$

reads

$$n_2^2 E^{\perp}_{lm2} - n_1^2 E^{\perp}_{lm1} = \frac{i\alpha\Theta_2}{k\pi}\hat{r}\cdot(\nabla\times\boldsymbol{E}_2) - \frac{i\alpha\Theta_1}{k\pi}\hat{r}\cdot(\nabla\times\boldsymbol{E}_1) \tag{72}$$

$$E^{\perp}_{lm2} - n^2 E^{\perp}_{lm1} = -\frac{i\alpha\Theta_2}{k\pi}\frac{l(l+1)}{r}E^{(2)}_{lm2} + \frac{i\alpha\Theta_1}{k\pi}\frac{l(l+1)}{r}E^{(2)}_{lm1}$$

$$= \frac{i\alpha}{k\pi}\frac{l(l+1)}{r}E^{(2)}_{lm}\Theta,$$

where the last line is obtained after inserting the continuity of the $E^{(2)}_{lm}$ components derived from the second boundary condition Eq. (70) and using $\Theta_2 = 0$ (topological trivial material outside of the sphere) and $\Theta_1 = \Theta$.

4. Boundary condition from the fourth Maxwell equation ($\hat{t}^{1,2}\,\widehat{=}\,$ tangent unit vector into $\boldsymbol{\Psi}_{lm}$ and $\boldsymbol{\Phi}_{lm}$ direction)

$$\frac{\alpha}{\pi}\left(\boldsymbol{t}^1\boldsymbol{E}_1\Theta_1 - \boldsymbol{t}^1\boldsymbol{E}_2\Theta_2\right) = \left(\boldsymbol{t}^1\boldsymbol{B}_1 - \boldsymbol{t}^1\boldsymbol{B}_2\right) \tag{73}$$

and hence

$$\frac{\alpha}{\pi}(E^{(1)}_{lm1}\Theta_1 - E^{(1)}_{lm2}\Theta_2) = -\frac{1}{ik}\left[\left(\frac{\mathrm{d}E^{(2)}_{lm1}}{\mathrm{d}r} + \frac{1}{r}E^{(2)}_{lm1}\right) - \left(\frac{\mathrm{d}E^{(2)}_{lm2}}{\mathrm{d}r} + \frac{1}{r}E^{(2)}_{lm2}\right)\right]. \tag{74}$$

Upon inserting Eq. (58) we obtain

$$\frac{\alpha r^2}{\pi l(l+1)}\left(\left(\frac{\mathrm{d}E^{\perp}_{lm1}}{r\mathrm{d}r} + \frac{2}{r^2}E^{\perp}_{lm1}\right)\Theta_1 - \left(\frac{\mathrm{d}E^{\perp}_{lm2}}{r\mathrm{d}r} + \frac{2}{r^2}E^{\perp}_{lm2}\right)\Theta_2\right)$$

$$= -\frac{1}{ik}\left[\left(\frac{\mathrm{d}E^{(2)}_{lm1}}{\mathrm{d}r} + \frac{1}{r}E^{(2)}_{lm1}\right) - \left(\frac{\mathrm{d}E^{(2)}_{lm2}}{\mathrm{d}r} + \frac{1}{r}E^{(2)}_{lm2}\right)\right]. \tag{75}$$

and finally

$$\frac{\alpha}{\pi l(l+1)}\left(\left(r\frac{\mathrm{d}E^{\perp}_{lm1}}{\mathrm{d}r} + 2E^{\perp}_{lm1}\right)\Theta_1 - \left(r\frac{\mathrm{d}E^{\perp}_{lm2}}{\mathrm{d}r} + 2E^{\perp}_{lm2}\right)\Theta_2\right)$$

$$= -\frac{1}{ik}\left[\left(\frac{\mathrm{d}E^{(2)}_{lm1}}{\mathrm{d}r}\right) - \left(\frac{\mathrm{d}E^{(2)}_{lm2}}{\mathrm{d}r}\right)\right], \tag{76}$$

or using BC1 Eq. (68)

$$-\frac{\alpha}{\pi l(l+1)}\left(r\frac{\mathrm{d}E_{lm}^{\perp}}{\mathrm{d}r}+2E_{lm}^{\perp}\right)\Theta=\frac{1}{ik}\left[\left(\frac{\mathrm{d}E_{lm1}^{(2)}}{\mathrm{d}r}\right)-\left(\frac{\mathrm{d}E_{lm2}^{(2)}}{\mathrm{d}r}\right)\right] \tag{77}$$

completing the 4 required BCs. Similarly to the field equations only 2 of the 4 BCs are affected by the topological term (i.e., those pertaining to the inhomogeneous Maxwell equations), while the other two remain as in the classical Mie theory.

We finally solve the system of BCs for the 6 coefficients $a_l$, $b_l$, $c_l$; the two free degrees of freedom (we only have 4 BCs) are fixed by the normalization (normal modes) or external boundary conditions (Mie scattering) as noted previously. Accordingly, the equation system solved for the normal modes in the main text (Eqs. (28)-(31))reads

$$\left(\begin{array}{cccccc|c} a_l^{(\perp)} & b_l^{(\perp)} & c_l^{(\perp)} & a_l^{(2)} & b_l^{(2)} & c_l^{(2)} & \\ \hline \frac{\mathrm{d}j_l^{\mathrm{i}}}{\mathrm{d}r}+\frac{j_l^{\mathrm{i}}}{r} & -\frac{\mathrm{d}j_l^{\mathrm{o}}}{\mathrm{d}r}-\frac{j_l^{\mathrm{o}}}{r} & -\frac{\mathrm{d}y_l}{\mathrm{d}r}-\frac{y_l}{r} & 0 & 0 & 0 & 0 \\ 0 & 0 & 0 & j_l^{\mathrm{i}} & -j_l^{\mathrm{o}} & -y_l & 0 \\ -n^2 j_l^{\mathrm{i}} & j_l^{\mathrm{o}} & y_l & -\frac{i\alpha l(l+1)\Theta}{k\pi}j_l^{\mathrm{i}} & 0 & 0 & 0 \\ \frac{ik\alpha\Theta}{\pi l(l+1)}\left(\frac{\mathrm{d}j_l^{\mathrm{i}}}{\mathrm{d}r}+\frac{j_l^{\mathrm{i}}}{r}\right) & 0 & 0 & \frac{\mathrm{d}j_l^{\mathrm{i}}}{\mathrm{d}r} & -\frac{\mathrm{d}j_l^{\mathrm{o}}}{\mathrm{d}r} & -\frac{\mathrm{d}y_l}{\mathrm{d}r} & 0 \end{array}\right). \tag{78}$$

The corresponding system for Mie scattering employing the Hankel function expansion reads

$$\left(\begin{array}{cc|cccc|c} a_l^{(2)} & a_l^{\perp} & d_l^{(2)} & e_l^{(2)} & d_l^{\perp} & e_l^{\perp} & \\ \hline y_l\frac{\mathrm{d}j_l^{\mathrm{i}}}{\mathrm{d}r}-j_l^{\mathrm{i}}\frac{\mathrm{d}y_l}{\mathrm{d}r} & \frac{\alpha\Theta ikR}{\pi l(l+1)}y_l\frac{\mathrm{d}(rj_l^{\mathrm{i}})}{r\mathrm{d}r} & -1 & -1 & 0 & 0 & 0 \\ -\frac{j_l^{\mathrm{o}}\frac{\mathrm{d}j_l^{\mathrm{i}}}{\mathrm{d}r}-j_l^{\mathrm{i}}\frac{\mathrm{d}j_l^{\mathrm{o}}}{\mathrm{d}r}}{i} & -\frac{\alpha\Theta kR}{\pi l(l+1)}j_l^{\mathrm{o}}\frac{\mathrm{d}(rj_l^{\mathrm{i}})}{r\mathrm{d}r} & -1 & 1 & 0 & 0 & 0 \\ -\frac{i\alpha l(l+1)\Theta}{\pi kR}j_l^{\mathrm{i}}\frac{\mathrm{d}(ry_l)}{r\mathrm{d}r} & y_l\frac{\mathrm{d}(rj_l^{\mathrm{i}})}{r\mathrm{d}r}-n^2 j_l^{\mathrm{i}}\frac{\mathrm{d}(ry_l)}{r\mathrm{d}r} & 0 & 0 & -1 & -1 & 0 \\ \frac{\alpha l(l+1)\Theta}{\pi kR}j_l^{\mathrm{i}}\frac{\mathrm{d}(rj_l^{\mathrm{o}})}{r\mathrm{d}r} & -\frac{j_l^{\mathrm{o}}\frac{\mathrm{d}(rj_l^{\mathrm{i}})}{r\mathrm{d}r}-n^2 j_l^{\mathrm{i}}\frac{\mathrm{d}(rj_l^{\mathrm{o}})}{r\mathrm{d}r}}{i} & 0 & 0 & -1 & 1 & 0 \end{array}\right). \tag{79}$$

The solution of the latter equation system (Eq. (32) and Eq. (33)) directly leads to the components of the scattering matrix $t_l$ defined by $d_l-e_l=t_l e_l$ (Eqs. (34)-(37)). The resulting complete expressions for the scattering cross section $\sigma_{\mathrm{s}}$ and extinction cross section $\sigma_{\mathrm{e}}$ than

reads:

$$
\sigma_s = \frac{2\pi}{k^2}\sum_{l=0}^{\infty}(2l+1)\left(\left(t_l^{11}+t_l^{12}\right)^2+\left(t_l^{21}+t_l^{22}\right)^2\right) \tag{80}
$$

$$
= \frac{2\pi}{k^2}\sum_{l=0}^{\infty}(2l+1)
$$

$$
\times \left[\left(\frac{\left(h_l^{(1)}\frac{\mathrm{d}\tilde{j}_l^{\mathrm{i}}}{\mathrm{d}\rho^{\mathrm{i}}}-j_l^{\mathrm{i}}\frac{\mathrm{d}\tilde{h}_l^{(1)}}{\mathrm{d}\rho^{\mathrm{o}}}\right)\left(j_l^{\mathrm{o}}\frac{\mathrm{d}\tilde{j}_l^{\mathrm{i}}}{\mathrm{d}\rho^{\mathrm{i}}}-n^2 j_l^{\mathrm{i}}\frac{\mathrm{d}\tilde{j}_l^{\mathrm{o}}}{\mathrm{d}\rho^{\mathrm{o}}}\right)-\frac{\alpha^2\Theta^2}{\pi^2}h_l^{(1)}\frac{\mathrm{d}\tilde{j}_l^{\mathrm{i}}}{\mathrm{d}\rho^{\mathrm{i}}}j_l^{\mathrm{i}}\frac{\mathrm{d}\tilde{j}_l^{\mathrm{o}}}{\mathrm{d}\rho^{\mathrm{o}}}}{\left(h_l^{(1)}\frac{\mathrm{d}\tilde{j}_l^{\mathrm{i}}}{\mathrm{d}\rho^{\mathrm{i}}}-j_l^{\mathrm{i}}\frac{\mathrm{d}\tilde{h}_l^{(1)}}{\mathrm{d}\rho^{\mathrm{o}}}\right)\left(h_l^{(1)}\frac{\mathrm{d}\tilde{j}_l^{\mathrm{i}}}{\mathrm{d}\rho^{\mathrm{i}}}-n^2 j_l^{\mathrm{i}}\frac{\mathrm{d}\tilde{h}_l^{(1)}}{\mathrm{d}\rho^{\mathrm{o}}}\right)-\frac{\alpha^2\Theta^2}{\pi^2}h_l^{(1)}\frac{\mathrm{d}\tilde{j}_l^{\mathrm{i}}}{\mathrm{d}\rho^{\mathrm{i}}}j_l^{\mathrm{i}}\frac{\mathrm{d}\tilde{h}_l^{(1)}}{\mathrm{d}\rho^{\mathrm{o}}}}\right.
$$

$$
\left.-\frac{\frac{\alpha l(l+1)\Theta}{\pi kR}j_l^{\mathrm{i}}\frac{\mathrm{d}\tilde{j}_l^{\mathrm{i}}}{\mathrm{d}\rho^{\mathrm{i}}}\left(\frac{\mathrm{d}\tilde{y}_l}{\mathrm{d}\rho^{\mathrm{o}}}j_l^{\mathrm{o}}-\frac{\mathrm{d}\tilde{j}_l^{\mathrm{o}}}{\mathrm{d}\rho^{\mathrm{o}}}y_l\right)}{\left(h_l^{(1)}\frac{\mathrm{d}\tilde{j}_l^{\mathrm{i}}}{\mathrm{d}\rho^{\mathrm{i}}}-j_l^{\mathrm{i}}\frac{\mathrm{d}\tilde{h}_l^{(1)}}{\mathrm{d}\rho^{\mathrm{o}}}\right)\left(h_l^{(1)}\frac{\mathrm{d}\tilde{j}_l^{\mathrm{i}}}{\mathrm{d}\rho^{\mathrm{i}}}-n^2 j_l^{\mathrm{i}}\frac{\mathrm{d}\tilde{h}_l^{(1)}}{\mathrm{d}\rho^{\mathrm{o}}}\right)-\frac{\alpha^2\Theta^2}{\pi^2}h_l^{(1)}\frac{\mathrm{d}\tilde{j}_l^{\mathrm{i}}}{\mathrm{d}\rho^{\mathrm{i}}}j_l^{\mathrm{i}}\frac{\mathrm{d}\tilde{h}_l^{(1)}}{\mathrm{d}\rho^{\mathrm{o}}}}\right)^2
$$

$$
+\left(\frac{\frac{\alpha\Theta kR}{\pi l(l+1)}j_l^{\mathrm{i}}\frac{\mathrm{d}\tilde{j}_l^{\mathrm{i}}}{\mathrm{d}\rho^{\mathrm{i}}}\left(j_l^{\mathrm{o}}\frac{\mathrm{d}\tilde{y}_l}{\mathrm{d}\rho^{\mathrm{o}}}-y_l\frac{\mathrm{d}\tilde{j}_l^{\mathrm{o}}}{\mathrm{d}\rho^{\mathrm{o}}}\right)}{\left(h_l^{(1)}\frac{\mathrm{d}\tilde{j}_l^{\mathrm{i}}}{\mathrm{d}\rho^{\mathrm{i}}}-j_l^{\mathrm{i}}\frac{\mathrm{d}\tilde{h}_l^{(1)}}{\mathrm{d}\rho^{\mathrm{o}}}\right)\left(h_l^{(1)}\frac{\mathrm{d}\tilde{j}_l^{\mathrm{i}}}{\mathrm{d}\rho^{\mathrm{i}}}-n^2 j_l^{\mathrm{i}}\frac{\mathrm{d}\tilde{h}_l^{(1)}}{\mathrm{d}\rho^{\mathrm{o}}}\right)-\frac{\alpha^2\Theta^2}{\pi^2}h_l^{(1)}\frac{\mathrm{d}\tilde{j}_l^{\mathrm{i}}}{\mathrm{d}\rho^{\mathrm{i}}}j_l^{\mathrm{i}}\frac{\mathrm{d}\tilde{h}_l^{(1)}}{\mathrm{d}\rho^{\mathrm{o}}}}\right.
$$

$$
+\left.\frac{\left(j_l^{\mathrm{o}}\frac{\mathrm{d}\tilde{j}_l^{\mathrm{i}}}{\mathrm{d}\rho^{\mathrm{i}}}-j_l^{\mathrm{i}}\frac{\mathrm{d}\tilde{j}_l^{\mathrm{o}}}{\mathrm{d}\rho^{\mathrm{o}}}\right)\left(h_l^{(1)}\frac{\mathrm{d}\tilde{j}_l^{\mathrm{i}}}{\mathrm{d}\rho^{\mathrm{i}}}-n^2 j_l^{\mathrm{i}}\frac{\mathrm{d}\tilde{h}_l^{(1)}}{\mathrm{d}\rho^{\mathrm{o}}}\right)-\frac{\alpha^2\Theta^2}{\pi^2}j_l^{\mathrm{o}}\frac{\mathrm{d}\tilde{j}_l^{\mathrm{i}}}{\mathrm{d}\rho^{\mathrm{i}}}j_l^{\mathrm{i}}\frac{\mathrm{d}\tilde{h}_l^{(1)}}{\mathrm{d}\rho^{\mathrm{o}}}}{\left(h_l^{(1)}\frac{\mathrm{d}\tilde{j}_l^{\mathrm{i}}}{\mathrm{d}\rho^{\mathrm{i}}}-j_l^{\mathrm{i}}\frac{\mathrm{d}\tilde{h}_l^{(1)}}{\mathrm{d}\rho^{\mathrm{o}}}\right)\left(h_l^{(1)}\frac{\mathrm{d}\tilde{j}_l^{\mathrm{i}}}{\mathrm{d}\rho^{\mathrm{i}}}-n^2 j_l^{\mathrm{i}}\frac{\mathrm{d}\tilde{h}_l^{(1)}}{\mathrm{d}\rho^{\mathrm{o}}}\right)-\frac{\alpha^2\Theta^2}{\pi^2}h_l^{(1)}\frac{\mathrm{d}\tilde{j}_l^{\mathrm{i}}}{\mathrm{d}\rho^{\mathrm{i}}}j_l^{\mathrm{i}}\frac{\mathrm{d}\tilde{h}_l^{(1)}}{\mathrm{d}\rho^{\mathrm{o}}}}\right)^2\right] \tag{81}
$$

and

$$
\sigma_e = \frac{2\pi}{k^2}\sum_{l=0}^{\infty}(2l+1)\,\Re\left\{t_l^{11}+t_l^{12}+t_l^{21}+t_l^{22}\right\} \tag{82}
$$

$$
= \frac{2\pi}{k^2}\sum_{l=0}^{\infty}(2l+1)
$$

$$
\times \,\Re\left\{\frac{\left(h_l^{(1)}\frac{\mathrm{d}\tilde{j}_l^{\mathrm{i}}}{\mathrm{d}\rho^{\mathrm{i}}}-j_l^{\mathrm{i}}\frac{\mathrm{d}\tilde{h}_l^{(1)}}{\mathrm{d}\rho^{\mathrm{o}}}\right)\left(j_l^{\mathrm{o}}\frac{\mathrm{d}\tilde{j}_l^{\mathrm{i}}}{\mathrm{d}\rho^{\mathrm{i}}}-n^2 j_l^{\mathrm{i}}\frac{\mathrm{d}\tilde{j}_l^{\mathrm{o}}}{\mathrm{d}\rho^{\mathrm{o}}}\right)-\frac{\alpha^2\Theta^2}{\pi^2}h_l^{(1)}\frac{\mathrm{d}\tilde{j}_l^{\mathrm{i}}}{\mathrm{d}\rho^{\mathrm{i}}}j_l^{\mathrm{i}}\frac{\mathrm{d}\tilde{j}_l^{\mathrm{o}}}{\mathrm{d}\rho^{\mathrm{o}}}}{\left(h_l^{(1)}\frac{\mathrm{d}\tilde{j}_l^{\mathrm{i}}}{\mathrm{d}\rho^{\mathrm{i}}}-j_l^{\mathrm{i}}\frac{\mathrm{d}\tilde{h}_l^{(1)}}{\mathrm{d}\rho^{\mathrm{o}}}\right)\left(h_l^{(1)}\frac{\mathrm{d}\tilde{j}_l^{\mathrm{i}}}{\mathrm{d}\rho^{\mathrm{i}}}-n^2 j_l^{\mathrm{i}}\frac{\mathrm{d}\tilde{h}_l^{(1)}}{\mathrm{d}\rho^{\mathrm{o}}}\right)-\frac{\alpha^2\Theta^2}{\pi^2}h_l^{(1)}\frac{\mathrm{d}\tilde{j}_l^{\mathrm{i}}}{\mathrm{d}\rho^{\mathrm{i}}}j_l^{\mathrm{i}}\frac{\mathrm{d}\tilde{h}_l^{(1)}}{\mathrm{d}\rho^{\mathrm{o}}}}\right.
$$

$$
-\frac{\frac{\alpha l(l+1)\Theta}{\pi kR}j_l^{\mathrm{i}}\frac{\mathrm{d}\tilde{j}_l^{\mathrm{i}}}{\mathrm{d}\rho^{\mathrm{i}}}\left(\frac{\mathrm{d}\tilde{y}_l}{\mathrm{d}\rho^{\mathrm{o}}}j_l^{\mathrm{o}}-\frac{\mathrm{d}\tilde{j}_l^{\mathrm{o}}}{\mathrm{d}\rho^{\mathrm{o}}}y_l\right)}{\left(h_l^{(1)}\frac{\mathrm{d}\tilde{j}_l^{\mathrm{i}}}{\mathrm{d}\rho^{\mathrm{i}}}-j_l^{\mathrm{i}}\frac{\mathrm{d}\tilde{h}_l^{(1)}}{\mathrm{d}\rho^{\mathrm{o}}}\right)\left(h_l^{(1)}\frac{\mathrm{d}\tilde{j}_l^{\mathrm{i}}}{\mathrm{d}\rho^{\mathrm{i}}}-n^2 j_l^{\mathrm{i}}\frac{\mathrm{d}\tilde{h}_l^{(1)}}{\mathrm{d}\rho^{\mathrm{o}}}\right)-\frac{\alpha^2\Theta^2}{\pi^2}h_l^{(1)}\frac{\mathrm{d}\tilde{j}_l^{\mathrm{i}}}{\mathrm{d}\rho^{\mathrm{i}}}j_l^{\mathrm{i}}\frac{\mathrm{d}\tilde{h}_l^{(1)}}{\mathrm{d}\rho^{\mathrm{o}}}}
$$

$$
+\frac{\frac{\alpha\Theta kR}{\pi l(l+1)}j_l^{\mathrm{i}}\frac{\mathrm{d}\tilde{j}_l^{\mathrm{i}}}{\mathrm{d}\rho^{\mathrm{i}}}\left(j_l^{\mathrm{o}}\frac{\mathrm{d}\tilde{y}_l}{\mathrm{d}\rho^{\mathrm{o}}}-y_l\frac{\mathrm{d}\tilde{j}_l^{\mathrm{o}}}{\mathrm{d}\rho^{\mathrm{o}}}\right)}{\left(h_l^{(1)}\frac{\mathrm{d}\tilde{j}_l^{\mathrm{i}}}{\mathrm{d}\rho^{\mathrm{i}}}-j_l^{\mathrm{i}}\frac{\mathrm{d}\tilde{h}_l^{(1)}}{\mathrm{d}\rho^{\mathrm{o}}}\right)\left(h_l^{(1)}\frac{\mathrm{d}\tilde{j}_l^{\mathrm{i}}}{\mathrm{d}\rho^{\mathrm{i}}}-n^2 j_l^{\mathrm{i}}\frac{\mathrm{d}\tilde{h}_l^{(1)}}{\mathrm{d}\rho^{\mathrm{o}}}\right)-\frac{\alpha^2\Theta^2}{\pi^2}h_l^{(1)}\frac{\mathrm{d}\tilde{j}_l^{\mathrm{i}}}{\mathrm{d}\rho^{\mathrm{i}}}j_l^{\mathrm{i}}\frac{\mathrm{d}\tilde{h}_l^{(1)}}{\mathrm{d}\rho^{\mathrm{o}}}}
$$

$$
+\left.\frac{\left(j_l^{\mathrm{o}}\frac{\mathrm{d}\tilde{j}_l^{\mathrm{i}}}{\mathrm{d}\rho^{\mathrm{i}}}-j_l^{\mathrm{i}}\frac{\mathrm{d}\tilde{j}_l^{\mathrm{o}}}{\mathrm{d}\rho^{\mathrm{o}}}\right)\left(h_l^{(1)}\frac{\mathrm{d}\tilde{j}_l^{\mathrm{i}}}{\mathrm{d}\rho^{\mathrm{i}}}-n^2 j_l^{\mathrm{i}}\frac{\mathrm{d}\tilde{h}_l^{(1)}}{\mathrm{d}\rho^{\mathrm{o}}}\right)-\frac{\alpha^2\Theta^2}{\pi^2}j_l^{\mathrm{o}}\frac{\mathrm{d}\tilde{j}_l^{\mathrm{i}}}{\mathrm{d}\rho^{\mathrm{i}}}j_l^{\mathrm{i}}\frac{\mathrm{d}\tilde{h}_l^{(1)}}{\mathrm{d}\rho^{\mathrm{o}}}}{\left(h_l^{(1)}\frac{\mathrm{d}\tilde{j}_l^{\mathrm{i}}}{\mathrm{d}\rho^{\mathrm{i}}}-j_l^{\mathrm{i}}\frac{\mathrm{d}\tilde{h}_l^{(1)}}{\mathrm{d}\rho^{\mathrm{o}}}\right)\left(h_l^{(1)}\frac{\mathrm{d}\tilde{j}_l^{\mathrm{i}}}{\mathrm{d}\rho^{\mathrm{i}}}-n^2 j_l^{\mathrm{i}}\frac{\mathrm{d}\tilde{h}_l^{(1)}}{\mathrm{d}\rho^{\mathrm{o}}}\right)-\frac{\alpha^2\Theta^2}{\pi^2}h_l^{(1)}\frac{\mathrm{d}\tilde{j}_l^{\mathrm{i}}}{\mathrm{d}\rho^{\mathrm{i}}}j_l^{\mathrm{i}}\frac{\mathrm{d}\tilde{h}_l^{(1)}}{\mathrm{d}\rho^{\mathrm{o}}}}\right\} \tag{83}
$$

These are the classical Mie theory results for the electric and magnetic mode (typically derived

by employing poloidal-toroidal decomposition techniques, see, e.g., [27]) modified by the axion term.

## A.3 Electron Scattering

Starting point for the analytical calculation of the axion-induced deflection of a focused electron beam at lateral coordinate $(b,0)$ into the $y-$direction (i.e., tangential to the spheres surface) is Eq. (46):

$$p_y^{\text{kin}}(b,0) = -\frac{1}{\pi v} \int_0^\infty d\omega \int dt \Re \left\{ e^{-i\frac{\omega}{v}z} \left( E_y^{\text{ind}}(b,0,z,\omega) + \frac{v}{c} B_x^{\text{ind}}(b,0,z,\omega) \right) \right\}. \quad (84)$$

In what follows, we split the calculation of the deflection into its electric and magnetic part. Moreover, we follow closely the derivation and notation of Ref. [35], which we adapted to our needs.

The axion-induced electric deflection into $y-$direction is given by the projection of the induced toroidal electrical field's $y-$component $E_y^{\text{m,ind}}$ at $(b,0)$, which is 0 in the topologically trivial case:

$$E_y^{\text{m,ind}}(\boldsymbol{r}) = E_{lm}^{(2),\text{ind}}(r)\Phi_{lm,y}(\boldsymbol{\Omega}) \quad (85)$$
$$= \sum_{l,m} d_{lm}^{(2)} h_l^{(1)}(kr)(r_z \partial_x - r_x \partial_z) Y_{lm}(\boldsymbol{\Omega}).$$

In the second step we inserted the radial dependency (outside of the sphere) of the outgoing toroidal solution and inserted the definition of the vector spherical harmonic field $\boldsymbol{\Phi} = \boldsymbol{r} \times \nabla Y_{lm}$. Moreover, the excitation coefficient has been denoted by $d_{lm}^{(2)}$ as in the main text. Now, the partial derivatives can be carried out yielding

$$E_y^{\text{m,ind}} = \frac{1}{2} \sum_{l,m} d_{lm}^{(2)} h_l^{(1)} \left( \sqrt{(l-m)(l+m+1)} Y_{lm+1} - \sqrt{(l+m)(l-m+1)} Y_{lm-1} \right). \quad (86)$$

Using the integral Eq. (97) from Ref. [35] the electrical part of the above projection integral can be evaluated to

$$p_y^E(b,0) = -\frac{1}{\pi} \int_0^\infty d\omega \int dt \Re \left\{ e^{-i\omega t} E_y^{\text{m,ind}}(b,0,vt,\omega) \right\} \quad (87)$$
$$= -\frac{1}{2\pi} \int_0^\infty d\omega \frac{1}{\omega} \sum_{l,m} \Re \left\{ d_{lm}^{(2)} i^{-1} \left( D_{lm}^+ - D_{lm}^- \right) \right\},$$

with

$$D_{lm}^+ = \sqrt{(l+m)(l-m+1)} A_{lm-1}^* K_{m-1}\left( \frac{\omega b}{v\gamma} \right), \quad (88)$$

$$D_{lm}^- = \sqrt{(l-m)(l+m+1)} A_{lm+1}^* K_{m+1}\left( \frac{\omega b}{v\gamma} \right). \quad (89)$$

In a final step we compute the excitation coefficient from the incoming field of the moving charge and the off-diagonal (axion) part of the transfer matrix

$$d_{lm}^{(2)} = t_l^{21} \frac{-2\pi i\omega}{c^2 \gamma} \frac{B_{lm}}{l(l+1)} K_m \quad (90)$$

yielding

$$p_y^E(b,0) = \frac{1}{c^2\gamma} \int_0^\infty d\omega \sum_{l,m} K_m \frac{B_{lm}}{l(l+1)} \left( \Re \left\{ D_{lm}^+ t_{l21} \right\} - \Re \left\{ D_{lm}^- t_{l21} \right\} \right), \quad (91)$$

where

$$B_{lm} = A_{lm+1}\sqrt{(l+m+1)(l-m)} - A_{lm-1}\sqrt{(l-m+1)(l+m)}. \tag{92}$$

The magnetic deflection into the $y$-direction is due to the poloidal magnetic field, which is obtained from the toroidal $\mathbf{E}^{\mathrm{m}}$ field according to

$$\mathbf{B}^{\mathrm{m}} = -\frac{i}{k}\nabla \times \mathbf{E}^{\mathrm{m}} \tag{93}$$

and hence

$$B_x^{\mathrm{m}} = -\frac{i}{k}\left(\partial_y E_z^{\mathrm{m}} - \partial_z E_y^{\mathrm{m}}\right). \tag{94}$$

The second term in the bracket can be directly derived from the electric deflection by partial integration of (87)

$$
\begin{aligned}
p_y^{B(2)}(b,0) &= \frac{1}{\pi c}\int_0^\infty \mathrm{d}\omega \int \mathrm{d}z\, \Re\left\{e^{-i\frac{\omega}{v}z}\frac{i}{k}\partial_z E_y^{\mathrm{m,ind}}\right\} \\
&= \frac{1}{c^2\gamma}\int_0^\infty \mathrm{d}\omega \sum_{l,m} K_m \frac{B_{lm}}{l(l+1)}\left(\Re\left\{D_{lm}^+ t_{l21}\right\} - \Re\left\{D_{lm}^- t_{l21}\right\}\right),
\end{aligned}
\tag{95}
$$

while the first term derives from the loss probability (Eq. (43)) taking into account that $\partial_y = \frac{1}{b}\partial_\varphi \to \frac{im}{b}$ at the locus of the beam $(b,0,z)$

$$
\begin{aligned}
p_y^{B(1)}(b,0) &= -\frac{1}{\pi c}\int_0^\infty \mathrm{d}\omega \int \mathrm{d}z\, \Re\left\{e^{-i\frac{\omega}{v}z}\frac{i}{k}\partial_y E_z^{\mathrm{m,ind}}(z,\omega)\right\} \\
&= \frac{1}{c^2 b}\sum_{l,m}\int_0^\infty \mathrm{d}\omega \frac{1}{\omega}K_m^2\left(\frac{\omega b}{v\gamma}\right)mC_{lm}^\perp \Re\{t_{l21}\} \\
&= 0.
\end{aligned}
\tag{96}
$$

This part of the deflection equates to zero as terms with positive and negative $m$ cancel each other. The above derivation made use of the analytical solution of the following integral

$$\int \mathrm{d}t\, e^{i\omega t}ih_l^{(1)}\left(\sqrt{k^2(b^2+(vt)^2)}\right)Y_{lm}^*\left(\mathrm{atan}\left(\frac{b}{vt}\right),0\right) = \frac{A_{lm}}{\omega}K_m\left(\frac{\omega b}{v\gamma}\right). \tag{97}$$

The definition of the $A_{lm}$ prefactors from Ref. [35] read

$$A_{lm} = \frac{1}{\beta^{l+1}}\sum_{j=m}^l \frac{C_j^{lm}}{\gamma^j}, \tag{98}$$

with

$$C_j^{lm} = \frac{i^{l-j}\alpha_{lm}(2l+1)!!}{2^j(l-j)!\frac{j-m}{2}!\frac{j+m}{2}!}I_{j,l-j}^{lm}, \tag{99}$$

$$\alpha_{lm} = \sqrt{\frac{2l+1}{4\pi}(l-m)!(l+m)!}, \tag{100}$$

and

$$I_{j,l-j}^{lm} = (-1)^m \int_{-1}^1 \mathrm{d}\mu(1-\mu^2)^{j/2}\mu^{l-j}P_l^m(\mu), \tag{101}$$

which may be solved analytically [35].

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
