# Peer review of "Axion Mie Theory of Electron Energy Loss Spectroscopy in Topological Insulators"

_SciPost Physics Core, doi:SciPost Phys. Core 4, 023 (2021)_

## Round 1 · Referee Report · Anonymous (Referee 1) · 2020-3-17

Report
What is new is that the advent of topological insulators has given us an excuse to renew studies of electrodynamics with an extra axion term in the Lagrangian, which leads to modified constitutive relations mixing quantities that are supposed to be magnetic or electric respectively. Many papers have been written redoing a lot of classic electrostatics and electrodynamics calculations with this extra term included: mirror charges, reflection off a plane, surface plasmons, ....
The modifications due to the axion term are often suppressed by two powers of the fine structure constant alpha and therefore difficult to observe. Some effects however are linear in alpha and, maybe more importantly, are corrections to quantities that would vanish without the axion term. The first, and most famous, such effect is the appearance of a magnetic monopole mirror charge at a planar interface.
The authors find, in this work, such an order alpha effect: a mixing of TE and TM modes in the scattering process off the sphere that can in principle be observed experimentally. This finding is in principle not surprising; exactly the same phenomenon has been observed previously for surface plasmons on a planar interface and it is the same mechanism that gives rise to this phenomenon here. But the careful generalization to the spherical geometry as well as the more detailed exposition of how one could experimentally confirm this effect make this work go substantially beyond the existing literature and make it worth while publishing.
One small modification I would like to see is a more careful discussion about the range of validity of the description of the TI in terms of the axion electrodynamics. Theorists love to use this description as it makes it easy to calculate. But this effective description is not always applicable (the surface modes need to be gapped and the effective description applies for energies below the gap), and given that the authors aim to connect to real experiments I would believe a somewhat more detailed discussion of the range of applicability of the action used would be desirable.

---

## Round 1 · Referee Report · Anonymous (Referee 2) · 2020-4-14

Strengths
- The authors have solved the problem of scattering by a sphere made of an isotropic dielectric characterized by a homogeneous relative permittivity and a homogeneous axionic constant.
Weaknesses
-
The solution of the homogeneous counterpart of Eq. (10) with spatially constant $\epsilon$ and $\Theta$ in the spherical coordinate system was first given by Bohren [ “Light scattering by an optically active sphere,” Chem. Phys. Lett. 29, 458-462 (1974)]. Just set $\alpha=\beta$ in the 1974 paper.
-
The Mie theory provided by the authors was completely formulated by Bohren in 1974. Just set $\alpha=\beta$ in the 1974 paper. This landmark paper went unnoticed by Ge et al. [“Electromagnetic scattering by sphere of topological insulators,” Opt. Commun. 354, 225-230 (2015)] and well as by the authors of the present manuscript.
-
There is another way to solve the same problem, whereby the axionic term does not enter constitutive relations but enters boundary conditions. That way for the Mie theory was adopted by Lakhtakia and Mackay ["Electromagnetic scattering by homogeneous, isotropic, dielectric-magnetic sphere with topologically insulating surface states,” J. Opt. Soc. Am. B 33, 603-609 (2016)]. Section 2.F of the 2016 paper provides a comparison of the two ways.
-
Whereas Bohren provided expressions only a for an incident plane wave, Lakhtakia and Mackay considered any source that lies outside the sphere. This is significant because the authors mention the importance of the “incorporation of arbitrary sources of fields” to justify their work in comparison to that of Ge et al. [Ref. 22].
-
The vector spherical wave functions of Barrera et al. [Ref. 23] are more cumbersome than those of Stratton [Electromagnetic Theory, McGraw-Hill, 1941] and Morse and Feshbach [Methods of Theoretical Physics Vol. II, McGraw-Hill, 1953]. Many of the manipulations presented by the authors in this manuscript would be eased if the vector spherical wave functions of Stratton and Morse&Feshbach are employed. Their orthogonalities on a unit sphere are exceedingly simple and lead to very simple manipulations, as the authors would find on consulting their Ref. 20. Neither Appendix A nor Appendix B are needed in 2020.
6a. Instead or TE and TM classification which is strictly valid as $r\to\infty$, a classification in terms of toroidal and poloidal fields is much better. because it applies everywhere. The electric field is poloidal if the magnetic field is toroidal (in isotropic achiral medium) and vice versa. This classification follows immediately from the vector spherical wave functions of Stratton and Morse&Feshbach; see Chandrasekhar and Kendall [“On force-free magnetic fields,” Astrophys. J. 126, 457-461 (1957)].
6b. The toroidal and poloidal classification will impact the dubious straightness of arrows drawn in Fig. 2 at distance of just 0.1 radius from the surface of the sphere.
7a. Since the distance $r$ is specified in nm in Figs. 1 and 2, this reviewer was surprised at not being able to find the frequency or the free-space wavelength used for the calculations. Neither was the relative permittivity reported for the material of the sphere nor the axionic constant.
7b. The same lacuna is evident for Fig. 3.
- The axionic admixing of TE and TM modes (as the authors put it) is trivial, all the more so because it is already known from the 1974 and the 2016 papers mentioned above. What is needed is a numerical estimate of a measurable quantity at a realizable frequency for a sphere made of a real material that has axionic properties. That is missing.
Report

---

## Round 2 · Referee Report · Anonymous · 2020-10-9

Report

I still think the discussion of the range of validity could contain more details, but the authors response is acceptable. Given the long lists of concerns the other referee had about this work, I believe that these issues are much more important than my more qualitative concerns. It appears that the authors addressed the other referee's comments in detail in their response, I will be curious to see what the other referee says. I have no objections anymore to publishing this article.

---

## Round 2 · Referee Report · Anonymous · 2020-10-19

Strengths

1. Possibly, the treatment of EELS.

Weaknesses

1. Claim of novelty for the "axionic Mie problem" is not valid, in view of Refs. 10, 11, and 13.
2. Suboptimal field representation when the optimal is known.
3. Numerical results are provided only for electrically small spheres, but not for larger spheres that would make the suboptimality apparent.

Report

In their response, the authors have stated that the work of Bohren (Ref. 11), Lakhtakia and Mackay (Ref. 10), and Ge et al. (Ref. 13) is applicable for the "homogeneous", whereas their Sec. 2 is novel because it deals with the "inhomogeneous" case.

The title of this paper is "Axion Mie theory for a spherical topological insulator". In addition, the axionic parameter $\Theta\in\{0,\pi\}$, according to the authors. Finally, they deal with regions with spatially constant constitutive parameters. So, the sphere is made of a material with $\Theta=\pi$ and the outside medium has $\Theta=0$. The sources lie outside the sphere. So long as the sources are not on the boundary, and are assumed to be unaffected by the scattered field, the theory of Refs. 10, 11, and 13 applies and suffices for the Mie scattering problem.

Even if fields in a region with spatially constant constitutive parameters are considered when that region has sources, those can be obtained from standard treatments since the dyadic Green function for such a medium is known, its singularity is known, and its bilinear expansion is known. This knowledge is a century old.

Following are detailed comments.

A. This reviewer obtained $\nabla\times\vec{B} =-(\alpha/\pi)\nabla\times(\Theta\vec{E})+ik\epsilon\vec{E}-(\alpha/\pi)ik\Theta\vec{B}+(4\pi/c)\vec{j}$ from Eq. (4). This will lead to $\nabla\times\vec{B} =-(\alpha/\pi)(\nabla\Theta)\times\vec{E}-(\alpha/\pi)\Theta\nabla\times \vec{E}
+ik\epsilon\vec{E} -(\alpha/\pi)ik\Theta\vec{B}+(4\pi/c)\vec{j}$ $=-(\alpha/\pi)(\nabla\Theta)\times\vec{E}-(\alpha/\pi)\Theta (\nabla\times \vec{E}+ik \vec{B})
+ik\epsilon\vec{E} +(4\pi/c)\vec{j}$ $=-(\alpha/\pi)(\nabla\Theta)\times\vec{E} +ik\epsilon\vec{E} +(4\pi/c)\vec{j}$ by virtue of Eq. (5).

Now, Eq. (5) also delivers $\nabla\times(\nabla\times\vec{E}) + ik (\nabla\times\vec{B})=0$.
Substitute the previously derived equation here to get
$\nabla\times(\nabla\times\vec{E}) -ik(\alpha/\pi)(\nabla\Theta)\times\vec{E}
-k^2\epsilon\vec{E}+i(4\pi k/c) \vec{j}
=0$. Check the signs of various terms in Eq. (15).

B. The authors have written that Eq. (16) holds for "spatially constant $\epsilon$". That
is incorrect. Equation (3) yields $\nabla\cdot\vec{E}= (\alpha/\pi\epsilon)\nabla\cdot(\Theta\vec{B})+
(4\pi/\epsilon)\rho
$ when $\epsilon$ is spatially constant. This equation can be simplified to
$\nabla\cdot\vec{E}= (\alpha/\pi\epsilon)(\nabla\Theta)\cdot \vec{B}+ (\alpha\Theta/\pi\epsilon)\nabla\cdot \vec{B}+(4\pi/\epsilon)\rho $
$= (\alpha/\pi\epsilon)(\nabla\Theta)\cdot \vec{B}+(4\pi/\epsilon)\rho $. There is no way Eq. (16) can be correct.

However, if both $\epsilon$ and $\Theta$ are spatially constant, then $\nabla\cdot\vec{E}=
(4\pi/\epsilon)\rho $ which leads to $\nabla(\nabla\cdot\vec{E})=
(4\pi/\epsilon)\nabla\rho $. Because $\nabla\times(\nabla\times\vec{E})=\nabla(\nabla\cdot\vec{E})-\nabla^2\vec{E}$, we get $\nabla\times(\nabla\times\vec{E})=(4\pi/\epsilon)\nabla\rho-\nabla^2\vec{E}$. Substitute in the corrected version of Eq. (15) then.

C. The authors have provided Eq. (17) with reference to Barrerra et al. [Eur. J. Phys. 6, 287 (1985)]. The authors claim that the vector spherical harmonics $\vec{Y}_{\ell m}(\vec{r})$, $\vec{\Psi}_{\ell m}(\vec{r})$, and $\vec{\Phi}_{\ell m}(\vec{r})$ "form a complete basis for vector fields". The authors need to provide an explicit proof for their claim with respect to the homogenous version of Eq. (16), because completeness has been skirted by Barrerra et al. as well as by the authors cited in that paper for the same issue.

D. In the 2nd paragraph of Sec. 2, the authors state that they "tackle the problem by a piecewise solution in regions of spatially constant $\epsilon$ and $\Theta$." As this stipulation holds for Eq. (17), the spatial dependences of $E_{\ell m}^\perp(\vec{r})$, $E_{\ell m}^{(1)}(\vec{r})$, and $E_{\ell m}^{(2)}(\vec{r})$, must come from the spatial dependences of the source terms $\vec{j}(\vec{r})$ and $\rho(\vec{r})$.

D.1. At points other than a source point, the field can be found since the relevant dyadic Green function is known; see: Chap. 2 of Faryad and Lakhtakia [Infinite-Space Dyadic Green Functions in Electromagnetism, IoP Books, 2018]. This is commonly done in the electromagnetics literature for transmitting antennas.

D.2. At a source point, the field can be found since the singular part of the relevant dyadic Green function is known; see: Yaghjian [Proc. IEEE 68, 248 (1980)]. This would not be needed for the Mie scattering problem anyway.

D.3. A convenient alternative to Eq. (17) in a source-free region
with both $\epsilon$ and $\Theta$ spatially constant is $\vec{E}(\vec{r})=\sum_{\kappa\in\{1,3\}} \sum_{\ell=0}^\infty$
$\sum_{m=-\ell}^{\ell}$ $[A_{\ell m\kappa} \vec{L}_{\ell m\kappa}(k'\vec{r})+B_{\ell m\kappa} \vec{M}_{\ell m\kappa}(k'\vec{r}) +C_{\ell m\kappa} \vec{N}_{\ell m\kappa}(k'\vec{r})]$, with $k'=k\sqrt{\epsilon}$. The vector spherical wavefunctions $\vec{L}_{\ell m\kappa}(k'\vec{r})$, $\vec{M}_{\ell m\kappa}(k'\vec{r})$, and $\vec{N}_{\ell m\kappa}(k'\vec{r})$, are provided on pp. 1865-1866 of Morse and Feshbach [Methods of Theoretical Physics, McGraw-Hill, 1953]. These functions use the spherical Bessel function $j_\ell(k'r)$ if $\kappa=1$ and the spherical Hankel function $h_\ell^{(\nu)}(k'r)$ if $\kappa=3$, with the index $\nu$ depending on the sign convention chosen for time-harmonic fields. Finding the space-independent coefficients $A_{\ell m\kappa}$, $B_{\ell m\kappa}$, and $C_{\ell m\kappa}$ is not at all a problem, since the bilinear expansion of the relevant dyadic Green function is known; see: Wood [Marconi Rev. 34, 149 (1971)] as an example. Alternatively, see: Wood [Reflector Antenna Analysis and Design, IEE Peter Peregrinus, 1980]. (It is not difficult to see that $A_{\ell m\kappa}\equiv 0$.) The expansion provided in this Comment is particularly convenient for the Mie scattering problem.

According to Comments D.1 and D.2, Eqs. (18)-(24) are unnecessary. They are certainly unnecessary for the Mie scattering problem.

E. So now to the Mie scattering problem. Suppose a sphere of radius $a$ is made of a homogeneous material with constitutive parameters $\epsilon\in\mathbb{C}$ and $\Theta=\pi$. The external medium extends to infinity in all directions and is the same as free space (i.e., $\epsilon=1$ and $\Theta=0$). The origin of the coordinate system lies at the center of the sphere. Suppose that all sources lie in the region $r>b>a$ and are unaffected by the scattered field. In the region $r<b$, the incident field will be given by the representation in Comment D.3 with $A_{\ell m\kappa}\equiv 0$, $B_{\ell m3}\equiv 0$, $C_{\ell m3}\equiv 0$, and $\epsilon=1$, as has been shown by Wood and is easy to show since the bilinear expansion of the free-space dyadic Green function is known. The scattered field will be given by the same representation with $A_{\ell m\kappa}\equiv 0$, $B_{\ell m1}\equiv 0$, $C_{\ell m1}\equiv 0$, and $\epsilon=1$. Representation of the field induced inside the sphere is already available from Refs. 10, 11, and 13. The boundary-value problem has already been solved in this general setting in Ref. 10. (Refs. 11 and 13 solve the problem for an incident plane wave.) Section 2 and Appendix B are unnecessary.

F. Section 2 and Appendix B are not only unnecessary, but they are also suboptimal. The functions $\vec{L}_{\ell m\kappa}(k'\vec{r})$, $\vec{M}_{\ell m\kappa}(k'\vec{r})$ and $\vec{N}_{\ell m\kappa}(k'\vec{r})$ are solutions of the vector Helmholtz equation for free space ($k'=k$) and a homogeneous isotropic dielectric medium ($k'\ne {k}$). The vector spherical harmonics $\vec{Y}_{\ell m}(\vec{r})$, $\vec{\Psi}_{\ell m}(\vec{r})$, and $\vec{\Phi}_{\ell m}(\vec{r})$ used in Eq. (17) are solutions of the vector Laplace equation, and therefore are necessarily suboptimal. Completeness of $\vec{L}_{\ell m\kappa}(k'\vec{r})$, $\vec{M}_{\ell m\kappa}(k'\vec{r})$ and $\vec{N}_{\ell m\kappa}(k'\vec{r})$ was explicitly proved by Aydin and Hizal [J. Math. Anal. Appl. 117, 428 (1986)].

G. The radius of the sphere is 50 nm in Figs. 1--3, whereas the free-space wavelength is 620 nm. As the radius is less than 10\% of the wavelength and $\vert\epsilon\vert=1$, this is an electrically small sphere. The authors need to try calculating for a sphere of radius 500 nm (or 5000 nm) and compare the computational resources needed for using Eq. (17) instead of the theory of Refs. 10, 11, and 13 to appreciate the suboptimality of Eq. (17).

H. In Figs. 1-3, the authors have set $\epsilon=-1$ at 620-nm free-space wavelength. What kind of a topological insulator has $\epsilon=-1$? Similarly, for Fig.~4, they have used $\epsilon=-0.5$. A thorough discussion is necessary along with plausible examples.

I. This reviewer was unable to find in the manuscript the sources responsible for the distributions presented in Figs. 1--3.

J. The poloidal-toroidal decomposition of applies in any source-free region in which $\epsilon$ and $\Theta$ are spatially constant. To see that look at Eq. (15). The third term on the left side and the only term on the right side will not exist. Furthermore, Eq. (3) would then simplify to $\nabla\cdot\vec{E}=0$. Appendix A is misleading.

K. The axionic admixing of TE and TM modes (as the authors put it) remains trivial, all the more so because it is already known from Refs. 10, 11, and 13 in the context of the Mie scattering problem.

---

## Round 2 · Author Response

We would like to thank both referees for reviewing our manuscript “Axion Mie theory for a spherical topological insulator”. We are grateful to both reviewers for raising a number of important questions. In order to address these questions, major revisions have been implemented in the manuscript, which is the reason for the rather long time required for resubmitting the manuscript. We inserted a completely new section containing analytic calculations of electron energy loss (EEL) spectra and axion induced beam deflections pertaining to an electron beam flying aloof a TI sphere. This amounts to performing an analytic solution of the inhomogeneous Mie scattering problem (i.e., in the presence of charges) for an experimentally relevant case (i.e., EELS). By doing so we also highlight the importance and relevance of the decomposition into a system of 3-component vector spherical harmonics in our paper, which deviates from the normally used 2-component vector spherical harmonics typically employed in Mie scattering theory. Indeed the latter is based on the poloidal-toroidal decomposition of transverse EM fields (as pointed out by Ref. [2]), which, however, completely describes the homogeneous (Mie) scattering only. We elaborate this point further in the revised version and also expanded our analysis of the homogeneous scattering case, including an explicit and exact expression of the scattering matrix previously missing (see Eqs. (34-39) in the revised version). Apart of this, we added further relevant references and corresponding discussions at many points. We also corrected a few typos and addressed stylistic issues. For more details on the specific changes made, see the separate responses to each referee. With this we request the editor and the referees to consider the current version of the manuscript for publication in SciPost.

Thank you, Johannes Schultz on behalf of the authors

Referee 1:

In the revised version, we extended the discussion of the range of validity of the axion theory in terms of topological insulators (TIs). We particularly elaborated on the significance of the gap and the magnitude of α.

Referee 2:

Before addressing the individual points, we would like to thank the referee for putting our work in the context of previous works, in particular for pointing out the important paper of Lakhtakia and Mackay.

Point 1: The solution of the homogeneous counterpart of Eq. (10) with spatially constant ϵ and Θ in the spherical coordinate system was first given by Bohren [ “Light scattering by an optically active sphere,” Chem. Phys. Lett. 29, 458-462 (1974)]. Just set α=β in the 1974 paper.

and

Point 2: The Mie theory provided by the authors was completely formulated by Bohren in 1974. Just set α=β in the 1974 paper. This landmark paper went unnoticed by Ge et al. [“Electromagnetic scattering by sphere of topological insulators,” Opt. Commun. 354, 225-230 (2015)] and well as by the authors of the present manuscript.

Our comment:
Thank you for pointing us at this important reference (Bohren in 1974), which is now referenced in the paper (Ref. [11] in the revised version). Similarly to our manuscript, Bohren introduced modified constitutive equations involving magnetic and electric field components (additional polarization depending on the magnetic flux density and additional magnetization depending on the electric field strength). However, only in the case of vanishing external sources do the constitutive equations used by us (Eqs. (7) and (8) in the revised version) reduce to those used by Bohren. In other words, in Bohren’s paper only the homogeneous case was treated, whereas our theory allows for external charges and currents, like the ones occurring in electron energy loss spectroscopy (EELS) experiments. As a consequence, our work provides an extension of Bohren’s 1974 paper.

Point 3: There is another way to solve the same problem, whereby the axionic term does not enter constitutive relations but enters boundary conditions. That way for the Mie theory was adopted by Lakhtakia and Mackay ["Electromagnetic scattering by homogeneous, isotropic, dielectric-magnetic sphere with topologically insulating surface states,” J. Opt. Soc. Am. B 33, 603-609 (2016)]. Section 2.F of the 2016 paper provides a comparison of the two ways.

Our comment:
The referee mentions an alternative way to formulate the problem, where the conducting surface states of the TI result in modified boundary conditions. Indeed, we also solved the problem using modified boundary conditions (Eqs. (21) - (24) in the revised version) derived from constitutive equations following from the axion-coupling. The modifications of the BCs are induced by the axionic terms in the constitutive relations (Eqs. (7) and (8) in the revised version). Consequently, the two approaches are identical and entail each other. Axionic constitutive equations affect observables by modified BCs and vice versa modified boundary conditions are consistent with an axion term in the Lagrangian density / axion terms in the constitutive equations on a more fundamental level. Summing up, both points of view are identical with respect to observable solutions (e.g., solutions of the Mie scattering problem in our case).

Point 4: Whereas Bohren provided expressions only a for an incident plane wave, Lakhtakia and Mackay considered any source that lies outside the sphere. This is significant because the authors mention the importance of the “incorporation of arbitrary sources of fields” to justify their work in comparison to that of Ge et al. [Ref. 22].

Our comment:
In our approach external sources are considered on a more general level in comparison to the work of Lakhtakia and Mackay. Indeed, Lakhtakia and Mackay first solve the homogeneous (i.e., transverse fields only) case and take care of external charges afterwards by adding transverse fields due to external sources to the homogeneous solution. In consequence this approach cannot treat cases, where charges / currents with correspondingly large longitudinal fields rest inside or very close to the sphere. Further away from the sphere, however, the restriction is less severe and the approach represents a good approximation. By employing 3-component vector spherical harmonics, we can indeed represent any 3D vector field (i.e., transverse and longitudinal) and hence solve the inhomogeneous differential equation without any approximations. This allows us, for instance, to model EELS experiments, where external charges and currents within the sphere are present (i.e., when the electron beam passes the sphere). Notwithstanding, in the newly added section containing explicit calculations of EEL spectra and electron beam deflection angles, we also focus on the aloof scattering geometry and neglect longitudinal solutions. The reason is that our analytical approach (containing special integrals, etc.) is only valid for transverse fields. We were not able to find a fully analytical solution including longitudinal fields (which of course does not preclude that an analytical solution may exist, let alone a numerical one).

Point 5: The vector spherical wave functions of Barrera et al. [Ref. 23] are more cumbersome than those of Stratton [Electromagnetic Theory, McGraw-Hill, 1941] and Morse and Feshbach [Methods of Theoretical Physics Vol. II, McGraw-Hill, 1953]. Many of the manipulations presented by the authors in this manuscript would be eased if the vector spherical wave functions of Stratton and Morse&Feshbach are employed. Their orthogonalities on a unit sphere are exceedingly simple and lead to very simple manipulations, as the authors would find on consulting their Ref. 20. Neither Appendix A nor Appendix B are needed in 2020.,

Point 6a: Instead or TE and TM classification which is strictly valid as r→∞, a classification in terms of toroidal and poloidal fields is much better. because it applies everywhere. The electric field is poloidal if the magnetic field is toroidal (in isotropic achiral medium) and vice versa. This classification follows immediately from the vector spherical wave functions of Stratton and Morse&Feshbach; see Chandrasekhar and Kendall [“On force-free magnetic fields,” Astrophys. J. 126, 457-461 (1957)].

and

Point 6b: The toroidal and poloidal classification will impact the dubious straightness of arrows drawn in Fig. 2 at distance of just 0.1 radius from the surface of the sphere.

Our comment:
The referee suggests to use vector spherical harmonics defined by Stratton [Electromagnetic Theory, McGraw-Hill, 1941] including poloidal and toroidal classification to simplify derivations. Indeed, the 2-component vector spherical harmonics of Stratton including the poloidal-toroidal classification is only valid for divergence free, i.e., transverse / solenoidal vector fields. The general hierarchy for the general decomposition of 3D vector fields (on compact domains) consists first of the Helmholtz decomposition into longitudinal / irrotational fields and transverse / solenoidal fields. The former may be represented by a scalar field, whereas the latter can be further decomposed into a poloidal and a toroidal field component (both can be represented by a scalar field). Thus (in a handwaiving way), we have a complete decomposition of a general 3D vector field into 3 scalar fields, which may be truncated to 2 components in the case of transverse fields only. In the present work, we explicitly allow for external charge densities in the Maxwell Equations, and thus neither the dielectric displacement nor the electric field strength is divergence free, and we have to resort to the complete 3-component decomposition by Barrera et al. (Ref. [28] in the revised version). This also explains why we used the TE and TM classification based on the general applicable Helmholtz-decomposition. In the current version, we make this point clearer and employ yet another popular naming convention from literature, where magnetic (electric) modes refer to TE (TM) modes respectively.

Point 7a: Since the distance r is specified in nm in Figs. 1 and 2, this reviewer was surprised at not being able to find the frequency or the free-space wavelength used for the calculations. Neither was the relative permittivity reported for the material of the sphere nor the axionic constant.

and

Point 7b: The same lacuna is evident for Fig. 3.

Our comment:
Thank you for bringing up that point. We have added the details of the calculations and apologize for any sloppiness.

Point 8: The axionic admixing of TE and TM modes (as the authors put it) is trivial, all the more so because it is already known from the 1974 and the 2016 papers mentioned above. What is needed is a numerical estimate of a measurable quantity at a realizable frequency for a sphere made of a real material that has axionic properties. That is missing.

Our comment:
As already pointed out, Bohren as well as Lakhtakia and Mackay solved only the homogeneous problem exactly. The effect of the axionic term can be derived neither from the solution of Bohren nor from Lakhtakia and Mackays work, if arbitrary external charges and currents are present. We, however, fully agree with the wish of the referee for explicit results on measurable quantities. In order to measure the signature of the axionic coupling, we propose EELS experiments. The high spatial and spectral resolution of modern EELS machines allows for the study of many different aspects (shape dependence, coupling, etc.) of localized surface resonances occurring in nanoparticles. In order to describe those EELS experiments theoretically, we explicitely considered arbitrary external charges and currents. For topologically non-trivial materials we found a unique deflection of the electron beam potentially measurable. In the revised version we derived analytic solutions of the EEL spectra and the electron beam deflection angles for a real scenario.

In summary we hope that we could convince the referee that the paper does fulfill the criteria regarding novelty, in view of our treatment of the general inhomogeneous case of Mie scattering at a spherical TI. We made major revisions in the paper to address the referee’s valuable comments. We have added in particular a new section containing explicit calculations of aloof electron scattering on a TI sphere pertaining to EELS.

---

## Round 2 · List of Changes

1. We inserted a completely new section containing analytic calculations of electron energy loss (EEL) spectra and axion induced beam deflections pertaining to an electron beam flying aloof a TI sphere.

2. We expanded our analysis of the homogeneous scattering case, including an explicit and exact expression of the scattering matrix previously missing (see Eqs. (34-39) in the revised version).

3. We extended the discussion of the range of validity of the axion theory in terms of topological insulators (TIs). We particularly elaborated on the significance of the gap and the magnitude of α.

4. We added further relevant references and corresponding discussions at many points, we corrected a few typos and addressed stylistic issues.

---

## Round 3 · Referee Report · Anonymous (Referee 5) · 2021-2-10

Report

After two reviews and two responses (considered unsatisfactory by this reviewer), this reviewer has come to the conclusion
that the only possible novelty (if any) in this manuscript is the “inhomogeneous case”. Is there a novelty? As far as this reviewer
can see, the problem handled by the authors is as follows:
(i) All fields are time harmonic.
(ii) The region $r < a$ is composed of a medium with constitutive scalar quantities $\epsilon_1$ and $\Theta_1$. Both quantities
are constant throughout the region $r < a$.
(iii) The region $r > a$ is composed of a medium with constitutive scalar quantities $\epsilon_2$ and $\Theta_2$. Both quantities
are constant throughout the region $r > a$.
(iv) All sources are confined to a region that lies outside the sphere $r ≤ a$ and is sufficiently distant from the sphere that it is
not affected by the scattered field.
All parts of the foregoing problem have been solved completely. Even if the sphere is replaced by some other 3D object, there
is no novelty left although data from the solution of specific boundary value problem could be of great interest to publish in
archival scientific journal.

Requested changes

The authors need to unequivocally state in the next revised version what the “inhomogeneous case” is. Furthermore, for the sake
of clarity, they need to remove the section containing Eqs. (25)-(39) so that the “homogeneous case” does not obscure any novelty
claimed by the authors for their work. (The “homogeneous TI sphere” problem has no novelty in it.) Then the revised version needs
further review.

---

## Round 3 · Author Response

We are again thankful for the referee’s valuable comments and criticism. The dispute surrounding the manuscript mainly circles around our claim to treat the inhomogeneous Mie problem in a way not presented elsewhere. Studying the references given in the referee's previous response, we still believe our results do represent a novel contribution. That pertains to the explicit derivation of the coupled radial differential equations (18)-(20) and the derivation of EELS cross sections and deflection angles, but also the explicit notation of the scattering matrix (34-37) of the homogeneous problem. The references given by the referee almost exclusively refer to the homogeneous problem and the projection of the inhomogeneous solution to transverse fields only (i.e., solutions of the homogeneous problem). In particular, they do not provide solutions at source points crucial for our aim to model EELS experiments being routinely carried out in our lab. As we did not and do not want to give the impression to rewrite Mie theory for topological insulators (in particular since the homogeneous problem has been discussed previously), however, we make the title of our paper more specific to the problem we are interested in: “Axion Mie Theory for Electron Energy Loss Spectroscopy in Topological Insulators”, in line with the comments of the referee.

Following are detailed comments pertaining to the points raised by the referee.

A.: Thank you for pointing out that sign mismatch. It has been corrected in the manuscript.

B.: Eq. (16) holds for spatially constant $\epsilon$ and $\Theta$, which resolves the issue (and was already noted a few lines below eq. (16) in the original version).

C.: We have indeed not proved the completeness of the vector spherical harmonics, which we, however, only use at the surface of the sphere (and not for decomposing fields in radial direction). However, there is a relatively straightforward way of obtaining this result via well known results from group theory. First note that the vector spherical harmonics (VSHs) we are using are simultaneous eigenfunctions of the operators ${\vec{J}}^2$ and $J_z$, where $\vec{J}=\vec{L}+\vec{S}$ is the total angular momentum adding the orbital angular momentum $\vec{L}$ to the spin operator $\vec{S}$ associated to a spin 1 particle (the photon). One of the main results of the general theory of angular momentum is that the simultaneous eigenstates of ${\vec{J}}^2$ and $J_z$ form a complete basis. Indeed, this basis is easily constructed using the addition theorems for angular momenta via the so called Clebsch-Gordan coefficients. The resulting VSHs have been used for more than eight decades in nuclear physics and other quantum mechanical problems [see, for instance, Appendix B of J. M. Blatt and V. F. Weisskopf, “Theoretical Nuclear Physics”, Springer-Verlag (1979), and V. B. Berestetskii, E. M. Lifshitz, and L. P. Pitaevskii, “Quantum Electrodynamics”, 2nd edition, Butterworth-Heinemann (2012), § 6 and § 7] . There is no issue regarding the completeness here, as it follows from group representation theory associated as applied to the construction of general angular momenta in quantum mechanics. The version used by Barrera et al. that we use is a mere rewriting of the VSH derived in the way described above. Indeed, these are precisely the same (up to proportionality factors), as the ones used in the textbook by Berestetskii et al., where a very lucid discussion is given.
Furthermore, they are specifically designed in such a way as to carry no radial dependency, i.e., the radial dependency in the expansion (17) is completely given by the expansion coefficients $E_{lm}^\perp$, $E_{lm}^{(1)}$, and $E_{lm}^{(2)}$. We hope that this resolves the issue.

D.: See point C., the spatial dependency of the expansion coefficients already stems from the fact that the expansion (17) is over the sphere and not full space, i.e., the radial dependency must be in the expansion coefficients (VSH are independent of $r$). Of course, source terms will also affect the spatial dependency, ultimately given by solution of (18-20).

D.1 and 2: We agree with the referee that the solution at any point in space (including source) may be found by appropriate Green’s functions. The crucial point is then to have explicit VSH expansions of the appropriate Green’s function / dyad. However, Chap. 2 of Faryad and Lakhtakia [Infinite-Space Dyadic Green Functions in Electromagnetism, IoP Books, 2018] as well as Wood [Marconi Rev. 34, 149 (1971)] / Wood [Reflector Antenna Analysis and Design, IEE Peter Peregrinus, 1980]) use only transverse functions (transverse VHS) in the expansion of the Green’s function, which then corresponds to the free space propagator of the fields. That is perfectly fine for the homogeneous Mie problem. However, if sources are present in the problem and we need to evaluate the fields at the sources (as is the case of electron energy loss spectroscopy), this type of approach amounts to an approximation at best. The accuracy of the latter ultimately depends on how close the sources (i.e., the electron beam) are to the sphere/nanoparticle. In EELS the beam can be also on the particle, which is the reason for writing (18-24) derived from Barrera VSH is the discussion of the fully inhomogeneous problem. The solution of (18-24) for a general source distribution represents the general solution to that problem. Alternatively, a complete expansion of the Green’s dyadics including the full set of vector spherical harmonics (i.e., including longitudinal fields) would also work. Yaghjian et al. indeed discusses general forms of Green’s dyadics but no expansion into vector spherical harmonics. Both Wood references (Wood [Marconi Rev. 34, 149 (1971)], Wood [Reflector Antenna Analysis and Design, IEE Peter Peregrinus, 1980]) also project out the longitudinal part. These references lack the complete expansion as required for the general solution of the inhomogeneous Mie scattering problem (which would certainly be especially useful for us).

D.3 We agree with the referee, however, that the homogeneous Mie problem is more conveniently solved by employing Morse and Feshbach VSHs, as these absorb the radial dependency of the fields into the vector spherical harmonics directly without requiring to explicitly calculate the radial dependency by solving a differential equation. Note, however, that even when considering only the homogeneous case a derivation employing a different set of bases (VSHs) can still have some additional merits as it provides another perspective on the problem.

E. Here we disagree. $A_{klm}=0$ is not correct for the described geometry. As the expansion of the source field is given by an integral over the whole space, there will be a longitudinal field in the presence of sources. One may restrict the problem to a finite sphere (excluding the source). In this particular case, however, surface terms show up in the corresponding Helmholtz decomposition, which reflect the presence of sources and modify the decomposition with respect to the homogeneous case. Moreover, we would not be able to compute the field at the source, which is, however, needed for the application case we have in mind. Only in the homogeneous case the longitudinal fields are exactly zero everywhere.

F. See above. We employed Barrera VSH to write the inhomogeneous problem as a system of 3 coupled inhomogeneous ordinary differential equations for the expansion coefficients. This would have been difficult with the Morse and Feshbach VSHs as they carry a radial dependency by themselves.

G. Following the suggestion of the referee, we calculated the field components for a 500 nm and a 5000 nm sphere without any significant increase of computation time. Note, that the calculated quantities (plotted in Fig. 1-3) are not a numerical solution of the inhomogeneous differential equations (18-20) but stemming from numerical calculations of the analytically derived expansion coefficients (eq. (25-27)) for the homogeneous case. Indeed, for the general inhomogeneous case (which also covers arbitrary external charges/currents and longitudinal field components) the differential equations (18-20) need to be solved numerically. Focusing on electron energy loss spectroscopy we explicitly need solutions for arbitrary external charges and currents (i.e., solutions at the position of the electron beam). Independent of numerical resources, the theory of Refs. 10, 11, and 14 does not provide solutions at source points, thus it cannot be applied as suggested by the referee.

H. Having plasmonic excitations on TIs in mind, we set the dielectric function negative (which is the fundamental condition for the presence of surface plasmonic excitations). One prominent TI with $\epsilon=-1$ at optical frequencies is Bi$_2$Se$_3$ (see e.g., [Talebi et al. ACS Nano (2016)]). A second example of a TI with negative dielectric function at optical frequencies is Bi$_2$Te$_3$ (see [Esslinger et al. ACS Photonics (2014)]). Both Bi$_2$Se$_3$ and Bi$_2$Te$_3$ are well known topological insulators. To make this clearer for the reader, we integrated a short discussion into the manuscript.

I. For more clarity we added additional information to the caption of Figs. 1-3. Note, that the plotted quantities are already denoted in the original manuscript (see e.g., the axis labels in Fig. 1).

J. We removed appendix A as it is not essential to the paper. Note, however, that the explicit form of the Helmholtz decomposition (and the ensuing toroidal-poloidal decomposition) depends on the space considered. If considering $R^3$ including sources, there is a non-vanishing longitudinal field. If considering some finite space, the decomposition requires simple connectivity and boundary terms must be considered.

K. We agree that the homogeneous problem has been treated previously, particularly in Ref. 10, see also our previous response. Note, however, that Refs. 10, 11 and 14 do not give the scattering matrix elements (34-37) explicitly, which to our point of view are furthermore not trivial! Indeed Ref. 14 even apply an additional approximation in neglecting all $\alpha^2$ terms. Whether such an additional approximation was also performed in Ref. 10 is not clear to us.

In conclusion, we addressed all the technicalities raised by the referee (e.g., concerning VSH or dielectric function) and hope that we could also explain better our point of view on the inhomogeneous Mie problem.

With best regards on behalf of the authors, Johannes Schultz

---

## Round 3 · List of Changes

1. We made the title of our paper more specific to the problem we are interested in and changed it to “Axion Mie Theory for Electron Energy Loss Spectroscopy in Topological Insulators”.

2. We revised the abstract.

3. We have included a brief discussion of plasmonic excitations on which we focus in the manuscript.

4. We added additional information to the caption of Figs. 1-3.

5. We removed appendix A as it is not essential to the paper.

6. Further relevant references and corresponding discussions have been added. In addition, we addressed some stylistic issues and typos.

---

## Round 4 · Author Response

Dear Dr. Attaccalite,
Unfortunately Prof. Javier García de Abajo has not accepted the invitation. He initially confirmed our request to review the manuscript, but he is currently not reachable. We have tried to recontact him directly multiple times, but unfortunately got no response. We have no explanation and hope he is doing well. Although we can not understand the referees criticism fully, we now tried to address the requested changes diligently (you can find a version in which changes to the manuscript have been marked red under https://seafile.ifw-dresden.de/f/9b5c16daec874984b302/). We added a more precise description of the inhomogeneous case as requested by the referee. Please note that we already referred to [1] in the previous version, which provides a detailed and unequivocal derivation of the inhomogeneous Mie problem (without axion contribution) and declaration of the applied approximations. Concerning the homogeneous problem, we have cited once more previous works on Mie scattering on TI spheres, and set our work more detailed in context to them. We have to reject, however, the request by the referee to remove the chapter containing Eqs. (25)-(39) for several reasons:
I) From our point of view the claim of the referee "(The “homogeneous TI sphere” problem has no novelty in it.)" is not correct. As pointed out already in the previous response letter there are several novelties in our treatment: - we provide an explicit and exact expression of the scattering matrix elements - we fully incorporate the axion term and do not apply a pertubation approach like e.g., Ge et al. [2] - we use a complete set of alternative vector spherical harmonics (VSH) which are, contrary to the claim of the referee, useful and necessary to solve the complete inhomogeneous problem, including arbitrary external charges and currents. The referee ignored our arguments completely and just claimed that all those aspects are not new without providing any references.
II) In condensed matter theory it is common practice to set novelties in context to previous works and recapitulate already treated points in order to make the subject more accessible to a broad audience of physicists. While the Mie theory is well known for the plasma physics and plasmonics community, it is less familiar to condensed matter physicists.
III) As already discussed in the previous version, the solution of the homogeneous equations are used to approximately solve the EELS problem ("one can directly adopt the results of the homogeneous calculations..."). Hence, the removal of Eqs. (25)-(39) would result in a less sound narrative regarding the derivation of the EELS solution.
In view of the critique by the referee on the lack of novelty, we elaborate further on the significance of the inhomogeneous solution and the role of the three-component set of VSH employed in the manuscript. The classical Mie approach considers scattering of transversal electromagnetic waves (i.e., in the absence of external charges and currents), which directly leads to a coupling of two components of the VSH. Consequently, only homogeneous transverse modes are possible and the solution space is restricted to such solutions, which can be exploited by using another set of strictly transverse two-component (poloidal and tororidal) VSHs, as advocated by the referee. To illustrate the lack of modes in case of free wave excitations, we performed boundary element simulations of the plasmonic response of a gold sphere with respect to a plane wave excitation, and an excitation by the evanescend field of a sharply focused electron beam, corresponding to the inhomogeneous case ( e.g., in EEL spectroscopy). Please see https://seafile.ifw-dresden.de/f/1cbb0bb930504473a9af/ for details and the resulting cross sections / spectra. One clearly observes that the mode below 0.5 eV only occurs in the full solution of the EELS problem and not in the homogeneous problem (i.e., plane wave excitation). Moreover, the low energy mode is also absent in the inhomogeneous Mie solution restricted to transverse mode solution space. Consequently, this results corroberates the importance of a complete three component set of VSH to solve the EELS problem completely. It also demonstrates that the restriction of the inhomogeneous case to transverse solutions correctly reproduces the transverse modes of the full solution, which we exploited in the dedicated EELS section of the paper. All in all we endeavored to address the criticized points and hope to fulfill now the criteria for publication in SciPost Core.
Faithfully yours, Axel Lubk
[1] F. G. de Abajo, Relativistic energy loss and induced photon emission in the interaction of a dielectric sphere with an external electron beam, Physical Review B 59(4), 3095 (1999)
[2] L. Ge, D. Han and J. Zi, Electromagnetic scattering by spheres of topological insulators, Optics Communications 354, 225 (2015)

---

## Round 4 · List of Changes

- We added a more precise description of the inhomogeneous case.
- We have cited previous works on Mie scattering on TI spheres, and set our work more detailed in context to them.

---

## Editorial Decision

published